# 🏛 OmniBench: Towards The Future of Universal Omni-Language Models

**Yizhi Li**[1,2]*, **Yinghao Ma**[1,3]*, **Ge Zhang**[1,4]*†, **Ruibin Yuan**[1,5], **Kang Zhu**[1,4], **Hangyu Guo**[1]
**Yiming Liang**[1], **Jiaheng Liu**[1,6], **Zekun Wang**[1,4], **Jian Yang**[1], **Siwei Wu**[1,2], **Xingwei Qu**[1,2]
**Jinjie Shi**[3], **Xinyue Zhang**[1], **Zhenzhu Yang**[1], **Yidan Wen**[1], **Yanghai Wang**[6], **Shihao Li**[6]
**Zhaoxiang Zhang**[6], **Zachary Liu**[7], **Emmanouil Benetos**[3], **Wenhao Huang**[1,4], **Chenghua Lin**[1,2]†
[1]M-A-P, [2]University of Manchester, [3]Queen Mary University of London, [4]01.ai
[5]Hongkong University of Science and Technology, [6]Nanjing University, [7]Dartmouth College

## Abstract

Recent advancements in multimodal large language models (MLLMs) have aimed to integrate and interpret data across diverse modalities. However, the capacity of these models to concurrently process and reason about multiple modalities remains underexplored, partly due to the lack of comprehensive modality-wise benchmarks. We introduce **OmniBench**, a novel benchmark designed to rigorously evaluate models' ability to recognize, interpret, and reason across **visual**, **acoustic**, and **textual** inputs simultaneously. We define language models capable of such tri-modal processing as the omni-language models (OLMs). OmniBench is distinguished by high-quality human annotations, ensuring that accurate responses require integrated understanding and reasoning across all three modalities. Our main findings reveal that: *i)* open-source OLMs exhibit critical limitations in instruction-following and reasoning capabilities within tri-modal contexts; and *ii)* most baselines models perform poorly (below 50% accuracy) even when provided with alternative textual representations of images or/and audio. These results suggest that the ability to construct a consistent context from text, image, and audio is often overlooked in existing MLLM training paradigms. To address this gap, we curate an instruction tuning dataset of 84.5K training samples, **OmniInstruct**, for training OLMs to adapt to tri-modal contexts. We advocate for future research to focus on developing more robust tri-modal integration techniques and training strategies to enhance OLMs. Codes, data and live leaderboard could be found at `https://m-a-p.ai/OmniBench`.

## 1 Introduction

The rapid advancement of artificial intelligence has ushered in a new era of multimodal large language models (MLLMs), capable of processing and interpreting diverse data types mainly involving images, audio, and text [Li and Lu, 2024]. These models aim to emulate human-like understanding of the world by integrating information across multiple sensory modalities and learning a comprehensive context from the environment. While significant strides have been made in developing MLLMs handling two of the modalities, the ability to concurrently process and reason about the three modalities remains a frontier yet to be fully explored.

The social impact of these MLLMs is far-reaching, providing transformative capabilities for a variety of domains. In healthcare, VLMs and ALMs have contributed to diagnosing [Liu et al., 2023a,

---

*Equal Contribution.
†Corresponding Authors.

39th Conference on Neural Information Processing Systems (NeurIPS 2025) Track on Datasets and Benchmarks.

Hemdan et al., 2023], and potentially combining *three modalities* [Meskó, 2023]. The integration of all vision, audio and text modalities is expected to significantly improve diagnostic accuracy and patient interaction, making healthcare more accessible and efficient. In urban environments, ALM can contribute to improving safety and traffic management by incorporating urban sound event detection during autonomous driving, such as recognizing audio of emergency vehicles and recognize their types or location with supplementary visual modality [Sun et al., 2021]. In addition, audio contributes to biodiversity monitoring [Terenzi et al., 2021, Liang et al., 2024a] and can be greatly enhanced by MLLM's ability to analyse both audio and video from a variety of sensors. Finally, it may help

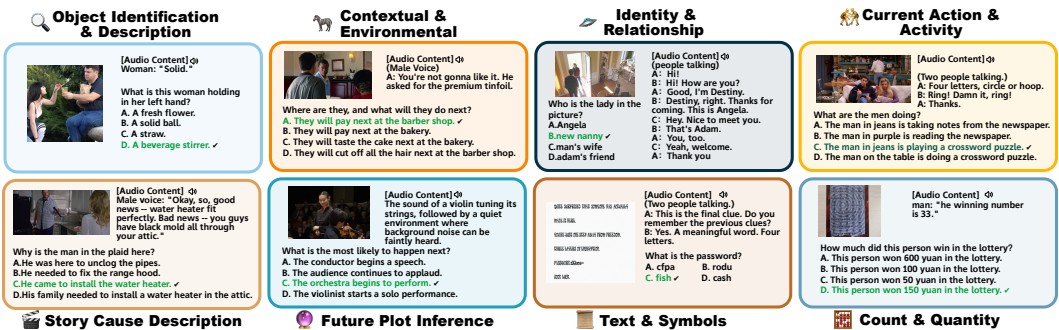

Figure 1: Example Data from Different Categories. The main contextual information is provided by the corresponding image and audio, while the question and options are expressed with text. Playable audio demos are available at the demo page.

robotics or LLM agents to provide better human-computer/robotic interaction (HCI/HRI) service in day-to-day life [Liang et al., 2024b, Su et al., 2023].

The challenge in advancing MLLMs lies not only in their development but also in our capacity to evaluate their performance comprehensively. Current benchmarks often solely focus on image or audios, or limited image-text [Yue et al., 2024, Zhang et al., 2024] or audio-text combinations [Yang et al., 2024] for the dual-modality vision-language models (VLMs) [Laurençon et al., 2024] or audio-language models (ALMs) [Chu et al., 2023a, Deng et al., 2023]. This gap in evaluation tools has hindered the community to assess and improve the holistic capabilities of models right before the dawn of general-purpose MLLMs.

To address this critical need, we introduce **OmniBench**, a pioneering universal multimodal benchmark designed to rigorously evaluate MLLMs' capability to recognize, interpret, and reason across visual, acoustic, and textual inputs *simultaneously*[3], which we define as the *omni-understanding and reasoning* ability of the omni-language models (**OLM**s) [Sun et al., 2024, Zhan et al., 2024, Lu et al., 2024b]. For instance, one can only derive the correct answer of the sample question in Figure 1 by: *1*) recognizing elements from the given image and audio to reconstruct the context; *2*) interpreting the semantics and relationships among the multimodal objects according to the textual instruction formed as question and options; *3*) reasoning and then answering with the complementary information from all the modalities. We distinguishes OmniBench by enforcing a unique constraint: accurate responses necessitate an integrated understanding and reasoning of *all multimodal contexts*. This approach ensures a more realistic and challenging assessment of multimodal large language models, mirroring the complex, interconnected nature of human cognition. To ensure the evaluation reliability, the development of OmniBench relies on high-quality human annotations. Furthermore, OmniBench additionally includes the answer rationales provided by the annotators to enhance the validity and ensure the benchmark aligned with human-level understanding.

Our initial findings using OmniBench reveal critical limitations in the omni-understanding capabilities of existing MLLMs:

- Although the existing open-source omni-language models have been trained with data in the three modalities, most of them can surpass the performance of random guess accuracy but sometimes hard to follow the instruction when provided with image and audio together in certain cases.
- In contrast, the proprietary models perform better overall but suffer from more accuracy drops when ablating the image or audio input.

---

[3]Note that domain-specific modalities from field like medical, biological or financial could also be included.

- Compared to models, the results of human evaluator show not only better overall performances but a distinct distribution (*e.g.*, much better on "Sound Event" audios and "Abstract Concept" tasks).
- In the context of using text as an alternative source of information to corresponding audio and images, the open-source VLMs and ALMs show relatively better results but remain in a preliminary level of capability to understand the given tri-modality context.

In the following sections, we 1) detail the data collection protocol of OmniBench; 2) present our evaluation results on current state-of-the-art MLLMs; 3) introduce the **OmniInstruct** dataset for omni-language model supervised fine-tuning; and 4) discuss the implications of our findings for the future of research and development.

## 2 Related Work

**Multimodal Large Language Models.** Recent advances in multimodal large language models have produced specialized encoders for audio processing [Radford et al., 2022, Chen et al., 2022, Li et al., 2023b, Wu et al., 2023b], which have been integrated into more comprehensive systems [Tang et al., 2023, Gong et al., 2023]. Notable progress in audio-focused dialogues has demonstrated promising capabilities in instruction-following and perception [Tang et al., 2023, Wang et al., 2023a, Wu et al., 2023a, Chu et al., 2023b]. In the visual domain, significant strides have been made through models that leverage pre-trained image encoders [Dosovitskiy, 2020, Touvron et al., 2020, Liu et al., 2021, Radford et al., 2021, Zhai et al., 2023]. These models have achieved success through visual-textual alignment, GPT-4 generated instruction data, and extensive pre-training [Li et al., 2023a, Liu et al., 2024b,a, Bai et al., 2023, Wang et al., 2023b, Young et al., 2024]. While most existing MLLMs focus on single-modality input processing, open-source models capable of processing multiple modalities generally show less competitive capabilities compared to closed-source alternatives. In this context, we define omni-language models (OLMs) as those capable of processing at least three different modalities simultaneously[4].

**Multimodal Understanding Benchmark.** Current vision-language benchmarks evaluate various capabilities including OCR, spatial awareness, multimodal information retrieval, and reasoning skills [Wu et al., 2024a, Yu et al., 2023, Liu et al., 2023c, Chen et al., 2024a, Yue et al., 2024, Zhang et al., 2024, Wu et al., 2024b]. These benchmarks assess models through multiple-choice tasks, complex vision-language problems, and multi-image relational understanding. In the audio domain, several benchmarks focus on speech recognition, audio QA tasks, and sound classification [Bu et al., 2017, Du et al., 2018, Panayotov et al., 2015, Drossos et al., 2020, Gong et al., 2022]. However, there remains a significant gap in comprehensive benchmarks that can assess models' ability to simultaneously process and integrate information from textual, audio, and visual inputs.

**Audio-Visual Understanding Datasets.** Existing audio-visual question answering datasets have primarily focused on object and sound identification [Yun et al., 2021, Li et al., 2022, Liu et al., 2024c, Yang et al., 2022]. While these datasets cover various aspects like temporal understanding and counting, they often lack comprehensive evaluation of causal inference and abstract reasoning. Some datasets don't require true multimodal integration for answering questions, as responses can often be deduced from a single modality. Recent work has attempted to address these limitations through human annotation and improved alignment between modalities [Chen et al., 2023, Gemmeke et al., 2017], but still lacks instruction-following evaluation capability. These limitations highlight the need for more comprehensive datasets that can effectively assess models' multimodal integration and reasoning abilities.

## 3 OmniBench

The OmniBench aims to create a comprehensive benchmark for evaluating multimodal large language models that support simultaneous image, audio, and text inputs. While OmniBench is designed to evaluate the understanding capability of MLLMs on cross-modality complementary information, the

---

[4]We target the models able to concurrently process image, audio, and text as a starting point since these are the most well-explored modalities in the field, but the "omni" concept is extendable.

Table 1: The Statistics of OmniBench Across Task Types. The word lengths of four options for each question are first averaged, and then the averages are calculated in group.

| Statistic | Causal Inference | | | (Temporal-)Spatial Entity | | Abstract Concept | | | |
|---|---|---|---|---|---|---|---|---|---|
| Sub-class of QA | Current Action & Activity | Story Description | Plot Inference | Object Identification & Description | Contextual & Environmental | Identity & Relationship | Text & Symbols | Count & Quantity | Overall |
| *Quantity* | | | | | | | | | |
| **Total** | 251 | 230 | 237 | 211 | 141 | 32 | 25 | 15 | 1142 |
| **Speech** | 78 | 182 | 179 | 162 | 104 | 31 | 22 | 13 | 771 |
| **Sound Event** | 147 | 27 | 37 | 28 | 25 | 1 | - | - | 265 |
| **Music** | 26 | 21 | 21 | 21 | 12 | - | 3 | 2 | 106 |
| *Word Length* | | | | | | | | | |
| **Question** | 4.68 | 5.75 | 7.47 | 7.00 | 6.85 | 6.22 | 7.32 | 8.72 | 6.25 |
| **Option** | 8.85 | 7.77 | 8.92 | 6.47 | 5.68 | 10.38 | 11.22 | 6.60 | 8.81 |
| **Img. Rationale** | 18.27 | 19.62 | 24.40 | 24.94 | 18.34 | 22.69 | 24.80 | 29.16 | 21.19 |
| **Audio Rationale** | 23.11 | 20.50 | 24.40 | 20.97 | 18.27 | 24.92 | 23.10 | 53.84 | 22.90 |
| **Audio Content** | 13.21 | 17.91 | 29.87 | 28.03 | 14.41 | 19.01 | 23.31 | 35.16 | 18.37 |
| *Multimodal Info.* | | | | | | | | | |
| **Img. Width** | 1283.75 | 1291.60 | 2394.93 | 1430.03 | 1141.39 | 1395.53 | 1338.51 | 1787.36 | 1322.36 |
| **Img. Height** | 842.32 | 776.11 | 2089.93 | 799.47 | 728.06 | 840.15 | 761.58 | 1168.04 | 818.64 |
| **Audio Len. (s)** | 7.35 | 9.82 | 11.22 | 11.43 | 8.03 | 8.63 | 11.43 | 15.63 | 9.22 |

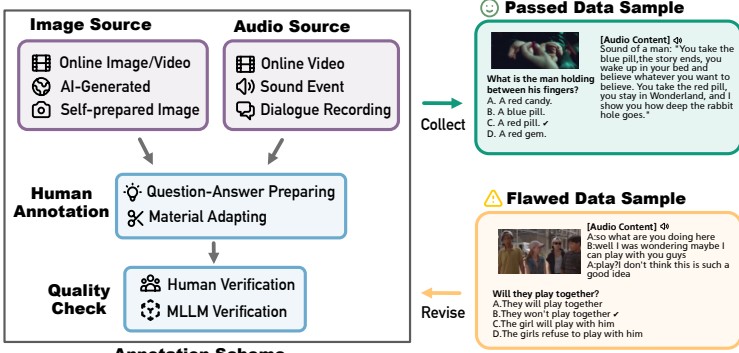

Figure 2: OmniBench Annotation Scheme. The annotation example shows flawed data that does not pass inspection because the information in audio *alone* is sufficient to answer. The audio in the flawed sample will then be sent back to annotators to edit.

models are required to interpret the multimodal input and provide accurate text answer. The problem could be formulated as following: given a tuple of (image, audio, text), the model is required to recognize the objects, re-build the contexts, and conduct reasoning based on the given information. The design logic and statistics of the dataset and the annotation protocols are introduced in this section.

## 3.1 Benchmark Design

Building on the foundation of existing multimodal benchmarks, our OmniBench introduces a refined taxonomy for task categorization that effectively captures a wide range of cognitive and reasoning abilities. Our framework organizes tasks into **three primary** categories: (1) *(temporal)-spatial entity*, which includes *Object Identification* for recognizing distinct entities and *Contextual & Environmental* for discerning the setting or backdrop of the events; (2) *causal inference*, comprised of *Story Cause Description* to infer narrative drivers, *Current Action & Activity* to understand ongoing dynamics, and *Future Plot and Purpose Inference* to anticipate subsequent developments; and (3) *abstract concept*, involving *Identity & Relationship* to identify and relate entities, *Text & Symbols* for symbolic interpretation, and *Count & Quantity* for numerical reasoning. This taxonomy is designed to evaluate both foundational perceptual skills and complex cognitive processes, thereby providing a comprehensive assessment of multimodal language models' (MLLMs) abilities to integrate and interpret diverse information sources. OmniBench includes **1142** question-answer pairs, with details on task types, text length, and the characteristics of images and audio presented in Table 1. It includes reasoning tasks involving: symbolic logic (e.g., matching audio cues to text clues), causal inference (e.g., predicting consequence given sound and scene), quantitative reasoning (e.g., counting musical or visual patterns), and commonsense reasoning (e.g., emotional mismatch or scene contradiction).

We also highlight instances of spatial-imagery reasoning (e.g., interpreting spatial layout from soundstage cues). While our current benchmark avoids explicit chain-of-thought traces, we believe it probes deep representational understanding across modalities. Though the benchmark contains 1,142 questions, they are intentionally diverse, and we argue the inter-sample heterogeneity justifies its use for statistical comparison. Appendix D outlines our bootstrapped hypothesis testing framework, which supports generalizable model comparisons within task types. The audio content of the dataset is categorized into speech, sound events, and music, enriching the diversity of stimuli for evaluating the models' tri-modal capabilities and aiding in the development of future omni-language models.

### 3.2 Annotation Protocol

**Annotation Scheme.** Our annotation scheme is built upon a fundamental principle: the correct answer to each question must require information from both the image and audio components. This ensures that the benchmark effectively evaluates the model's ability to analyze information across modalities. As shown in Figure 2, we implemented a rigorous annotation pipeline consisting of three stages: initial annotation, human inspection, and model inspection. Data samples that failed to meet our criteria at any stage were returned to annotators for revision, ensuring high-quality, multimodal-dependent samples. Through the whole process, 16 *annotators and* 5 *quality inspectors* are involved, all are full-time industrial data annotation employee with higher education backgrounds.

The questions are formalized as multi-choice question-answering (MCQ) but try to maintain a consistent logic that suggests the only one possible and accurate answer, *i.e.*, they can be potentially further re-organized into blank filling questions. Furthermore, when constructing the options, the annotators need to ensure at least one confusing wrong option. To ensure question difficulty, annotators were required to verify that questions and options were not trivially easy, lacked distinguishable patterns, and could not be answered by state-of-the-art MLLMs using image information alone. GPT-4 are allowed to use to provide initial annotator self-assessments of question quality. We restrict the images with a minimum resolution of 480P (854x480 pixels) and audio clips with a maximum duration of 30 seconds.

We implemented strict measures to maintain diversity across the dataset. This includes varying image and audio sources, limiting the frequency of individual speakers in audio clips to no more than five occurrences, and restricting the replication of similar instructions or questions. For instance, questions about specific environmental contexts were limited up to three samples. Importantly, annotators were required to provide audio transcripts (or music captions) and **rationales** for correct answers, detailing the specific information that should be derived from the image and audio modalities respectively. This approach not only aided in quality inspection but also laid the groundwork for future fine-grained evaluation.

**Quality Control.** Our quality control process was two-fold, including human inspection round and automatic inspection round assisted by MLLM. First, all annotated instruction-response pairs undergo cross-inspection by human annotators. Inspectors provide detailed reasons for any samples that failed to meet our stringent criteria, allowing for targeted revisions. Samples that pass human inspection are then subjected to a secondary check using a vision-language model LLaVA-1.6-34B [Liu et al., 2024a], where the the automatic quality inspection model is selected by considering trade-off between efficiency and performance. This automated process evaluates each sample under various ablation settings: *image and text only*, *audio transcript and text only*, and *text only* (repeated three times). Samples are only accepted if the model either rejected the task or made mistakes under these limited-

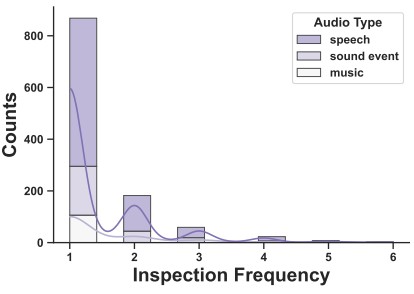

Figure 3: The Distribution of Inspection Frequency of the Passed Samples in OmniBench.

information scenarios, confirming the necessity of both visual and auditory information for correct responses. We plot the distribution of the inspection frequency of the passed samples in Figure 3, where we could find that 76% (868) of the passed samples do not require further modification under a well-defined annotation framework and 21.1% of them requiring 1-2 times of revision. During iterative quality checking, 9.58% (121) QA pairs are defined as "hard to recycle" and dumped.

## 3.3 OmniInstruct

To improve the model capability of tri-modal reasoning, we develop the **OmniInstruct** dataset to facilitate the supervised fine-tuning of models. This dataset leverages the following data sources: the MSRVTT-QA [Xu et al., 2017], AVQA [Yang et al., 2022] and Music-AVQA2.0[Liu et al., 2024c], all of which contain visual, audio and corresponding QA text resources. MSRVTT-QA and AVQA consist of short video clips, typically ranging from 10 to 20 seconds, with minimal scene changes, and music-AVQA 2.0 dataset include 1 minute music performance video. We only adopt the train and validation split of this dataset and regard the whole OmniBench as the test set of the task.

Table 2: Summarization of OmniInstruct Data Distribution across Modalities and Question Types.

|  | image | audio | how many | what | who | when&where | others |
|---|---|---|---|---|---|---|---|
| Train | 84,580 | - | 1,131 | 56,168 | 22,449 | 959 | 3,873 |
| Valid | 11,525 | - | 80 | 4,498 | 1,884 | 66 | 4,997 |
| Total | 96,105 | - | 1,211 | 60,666 | 24,333 | 1,025 | 8,870 |

To construct a dataset that aligns with the challenges proposed in OmniBench, we enhanced each question to connect with an audio track and an image extracted from the corresponding video and filter it with VLMs for better quality. Notably, we avoid the first and last five frames of each video to exclude transitional or obscure incomplete scenes that might distort the task's focus. For MSRVTT-QA train and valid subset, we discard videos without audio tracks and retain a dataset comprised 6,176 videos from the original set that include audio tracks alongside 151.7k QA pairs directly related to these videos. Then we use InternVL-2-76B to filter the questions from the three aforementioned datasets to *delete* (1) questions that can be answered only with an image, (2) questions irrelevant with image, potentially answerable only with audio, and (3) ambiguous or non-logical questions, where the detailed prompt and statistics could be found at Figure 5 and Table 9, Appendix B. After the processing pipeline, only 96k data samples remain for training and validation.

# 4 Experiment Settings

**Baseline Systems**   We select three groups of MLLM baselines according to the modalities available: *(i) omni-language models*: MIO-Instruct [Wang et al., 2024b], AnyGPT [Zhan et al., 2024], Video-SALMONN [Sun et al., 2024], UnifiedIO2 series [Lu et al., 2024b], the VITA series [Fu et al., 2024, 2025], OpenOmni Luo et al. [2025], Baichuan-Omni-1.5 Li et al. [2025], Qwen-2.5-Omni Xu et al. [2025]; *(ii) vision-language models*: InternVL-2 series [Chen et al., 2024b], Qwen2-VL series [Wang et al., 2024a], Deepseek-VL [Lu et al., 2024a], LLaVA-One-Vision series [Li et al., 2024], Cambrian series [Tong et al., 2024], Xcomposer2-4KHD [Dong et al., 2024], Idefics2 [Laurençon et al., 2024] as well as the derived Mantis-Idefics2 [Jiang et al., 2024]; *(iii) audio-language models*: LTU series [Gong et al., 2023], Mu-LLaMA [Liu et al., 2023b], MusiLingo [Deng et al., 2023], Qwen-Audio series [Chu et al., 2023a], SALMONN-Audio [Sun et al., 2024] and Audio-Flamingo [Kong et al., 2024]. We also include the API calls from proprietary models that could support image-text or audio-text inputs, including GPT4-o, Gemini Pro, Reka and Claude-3.5-Sonnet [Achiam et al., 2023, Team et al., 2023, Ormazabal et al., 2024, Anthropic, 2024]. We do not conclude them as in the group of VLMs, ALMs or OLMs (even not a single model) in our context at the moment since the mechanisms behind these models are not revealed [5]. Besides, we invite three musician with higher education background to test on the benchmark and use the average accuracy as a human expert baseline.

**Omni-Understanding Evaluation.**   The main focus of OmniBench is to evaluate how well could the MLLMs understand and reconstruct the context given information from image ($I$), audio ($A$) and text ($T$) modalities. Setting up questions with four available options for the models, we use accuracy, *i.e.*, the ratio matched letter of the correct option and model response, as the evaluation metric (*n.b.*, the accuracy of a random guess model is $25\%$ under this setting). Additionally, we test the models in an ablation setting of removing one of the image or audio inputs to further reveal a more comprehensive reasoning capability of the baselines and verify the robustness of our benchmark.

---

[5]The authors conclude from an investigation on September 22, 2024.

**Textual Approximation of Image and Audio.** For most of the existing MLLMs that only support two input modalities (($I, T$) or ($A, T$)), we build up a simulated evaluation setting allowing us to explore the potential of these models to become omni-language models in the future. We use the audio transcript ($A'$) annotated by human as the alternative of the audios to enable the evaluation on vision-language models. Regarding the audio-language models, we generate high-quality detailed captions of images ($I'$) automatically with a state-of-the-art VLM, InternVL-2-76B. In such an approximated evaluation setting, models go through the same process of inference and metric calculation as the vanilla one with textual alternatives of images or audios.

Table 3: Overall Omni-Undesratnding Results on Baseline Omni-Language Models. The overall (Image & Audio), image-ablated and audio-ablated results on all samples are provided.

| Input Context | Image & Audio | Audio | Image |
|---|---|---|---|
| MIO-Instruct (7B) | 24.80% | 25.39% | 27.93% |
| AnyGPT (7B) | 18.04% | 16.20% | 20.05% |
| video-SALMONN (13B) | 35.64% | **35.90%** | **34.94%** |
| VITA (8 × 7B) | 33.10% | 27.06% | 33.54% |
| VITA-1.5 (7B) | 33.40% | - | - |
| UnifiedIO2-large (1.1B) | 27.06% | 29.07% | 29.07% |
| UnifiedIO2-xlarge (3.2B) | 38.00% | 31.17% | 34.76% |
| UnifiedIO2-xxlarge (6.8B) | 33.98% | 32.49% | 33.45% |
| OpenOmni | 37.40% | - | - |
| Baichuan-Omni-1.5 | 42.90% | - | - |
| MiniCPM-o 2.6 | 40.50% | - | - |
| Qwen-2.5-Omni | **56.13%** | - | - |
| Gemini-1.5-Pro | 42.91% | 27.93% | 26.09% |
| Reka-core-20240501 | 30.39% | 23.12% | 30.65% |
| Human Evaluator | 63.19% | - | - |
| Human Expert | 74.03% | - | - |

# 5 Findings

We evaluate the selected baseline systems in multiple meticulously designed settings, and provide insights on the development of OLMs based on the results. All the claims stated in this section are supported by statistic significance verification, for more information please refer to Appendix D.

## 5.1 Results on Omni-Language Models

**Overall** Table 3 demonstrates that most open-source OLM baselines surpass random guessing accuracy across various setting, and the latest developed one (Qwen-2.5-Omni) could even outperform the proprietary models. Notably, the UnifiedIO2 series demonstrates inconsistent performance scaling with model size, indicating challenges in effectively leveraging increased capacity for multimodal understanding. In contrast, Gemini-1.5-Pro and Reka-core-20240501, the two available proprietary models evaluated in this tri-modal setting, shows more promising results. Regarding the scores across audio types, the Gemini-1.5-Pro shows a more balanced performances while Reka-core-20240501 showing

Table 4: OLM Baselines Overall Results Grouped by Audio Type.

| Model | Speech | Sound Event | Music |
|---|---|---|---|
| AnyGPT (7B) | 17.77% | 20.75% | 13.21% |
| Video-SALMONN (13B) | 34.11% | 31.70% | **56.60%** |
| VITA (8 × 7B) | 31.52% | 32.45% | 46.23% |
| UnifiedIO2-large (1.1B) | 25.94% | 29.06% | 30.19% |
| UnifiedIO2-xlarge (3.2B) | 39.56% | 36.98% | 29.25% |
| UnifiedIO2-xxlarge (6.8B) | 34.24% | 36.98% | 24.53% |
| Qwen-2.5-Omni | **55.25%** | **60.00%** | 52.83% |
| Gemini-1.5-Pro | 42.67% | 42.26% | 46.23% |
| Reka-core-20240501 | 31.52% | 26.04% | 33.02% |
| Human Evaluator | 58.71% | 75.85% | 64.15% |

a lag on modeling the sound events. Moreover, the comparison of Gemini-1.5-Pro's performance across full input context and ablated settings (image-removed and audio-removed) suggests that it effectively leverages information from all modalities to enhance its reasoning capabilities. While it demonstrates superior overall performance and balanced accuracy across audio types compared to most of the open-source alternatives, its accuracy remains below 50%. OmniBench's best-performing model (Qwen-Omni) also leads in audio reasoning. Interestingly, it combines separate audio and vision instruction tuning, suggesting that fusing different bimodal corpora can generalize to tri-modal

settings. Although most top-performing models still rely on modality-specific encoders, emerging models like AnyGPT—which use a universal tokenizer for multiple modalities—represent a more integrated architecture. We hypothesize that GPT-4o may follow this direction, and believe such designs will be necessary to scale OLMs to more complex understanding and generation tasks.

**Breakdown Results.** We present the breakdown of the performance of open-source omni-language model baselines across different audio types and task categories in the OmniBench evaluation in Table 4 and Table 5. The results reveal inconsistent performance patterns across audio types. For instance, despite overall poor performance, open-source baselines generally exhibit higher accuracy on speech audio, indicating a potential bias towards speech data. Besides, Video-Salmonn and Gemini-1.5-Pro provide better results on music subsets compared to their performance on speech and music, potentially due to their large corpus of music videos, though the music ethics of training foundation models are still under-discussion [Ma et al., 2024]. Across task categories, many of the

Table 5: OLM Baselines Overall Results Grouped by Task Category.

| Accuracy ↑ | Causal Inference | | | (Temporal-)Spatial Entity | | Abstract Concept | | |
|---|---|---|---|---|---|---|---|---|
| Sub-class of QA | Action & Activity | Story Description | Plot Inference | Object Identification & Description | Contextual & Environmental | Identity & Relationship | Text & Symbols | Count & Quantity |
| AnyGPT (7B) | 19.52% | 16.52% | 14.77% | 22.27% | 15.60% | 21.88% | 12.00% | 33.33% |
| Video-SALMONN (13B) | 31.47% | 28.26% | 25.74% | 62.56% | 36.88% | **37.50%** | 20.00% | 6.67% |
| VITA (8 × 7B) | 35.86% | 33.04% | 29.54% | 33.65% | 41.84% | 21.88% | 16.00% | 6.67% |
| UnifiedIO2-large (1.1B) | 29.88% | 20.87% | 31.65% | 30.81% | 23.40% | 18.75% | 24.00% | 6.67% |
| UnifiedIO2-xlarge (3.2B) | 32.27% | **33.48%** | 31.65% | **63.03%** | 34.04% | 34.38% | 24.00% | 20.00% |
| UnifiedIO2-xxlarge (6.8B) | 32.27% | 29.13% | 29.96% | 48.82% | 34.75% | 25.00% | 8.00% | **46.67%** |
| Gemini-1.5-Pro | **41.83%** | 30.87% | **32.91%** | 62.56% | **60.28%** | 31.25% | **28.00%** | 13.33% |
| Reka-core-20240501 | 25.50% | 24.78% | 20.68% | 49.76% | 39.01% | 28.12% | **28.00%** | 6.67% |
| Human Evaluator | 72.91% | 55.51% | 53.87% | 71.41% | 65.48% | 59.38% | 45.33% | 66.67% |

models, such as Gemini-1.5-Pro, tend to perform better on object identification and description tasks while struggling with more reasoning tasks such as plot inference and story description. This might be because visual entity recognition task is an essential component for image captioning and other type of pre-training dataset. Furthermore, some models like Gemini-1.5-Pro, Reka-core-20240501, and Video-SALMONN perform significantly badly on quantity & counting tasks compared with the rest of the tasks. But scaling up of UnifiedIO model contributes a lot to this type of task.

**Results on Music-related Questions** Model underperform on music subset in Table 4 and Table 7 is notable and multifactorial. While OmniBench is carefully curated to balance genre and linguistic diversity (e.g., avoiding overrepresentation of English pop), many audio-capable LLMs suffer from training-data bias. For instance, Qwen2-Audio leverages MTG-Jamendo for music tagging, yielding strong results there but poor generalization to other datasets like MTT [Ma et al., 2025]. In contrast to speech, music is more restricted by copyright and expensive to annotate for supervised tasks.

**Human Evaluation** We invite three human annotators to evaluate on the OmniBench questions, deriving a much higher accuracy $63.19\%$ compared to all the OLMs[6]. The human evaluator results hold a Fleiss' Kappa value of $0.421$ suggests, suggesting a high level of inter-annotator agreement. Different from models, human evaluation shows a particularly good performance on "Sound Event" among the audio types, which is reasonable since the sounds are short and straightforward without requiring complex reasoning to recognize. Moreover, human shows higher scores on "Abstract Concept" tasks, potentially because these tasks require more reasoning.

## 5.2 The Effectiveness of OmniInstruct

To validate the effectiveness of OmniInstruct, we conducted experiments using MIO-instruct, a 7B parameter omni-language model previously trained only on audio-language and vision-language pairs. We evaluated the impact of supervised fine-tuning using OmniInstruct through two experimental settings. First, we created a compact subset of OmniInstruct containing 6.4K samples (approximately

---

[6]Human performance is limited by time constraints, annotators' non-native language or music knowledge gaps, and their associate-level education.

7.5% of the full dataset) to enable efficient experimentation. In our first setting (MIO-Instruct-Omni-V1), we directly fine-tuned the model using its vanilla multimodal tokenizers. In the second setting (MIO-Instruct-Omni-V1-voice-filtered), we refined the approach by filtering out non-speech tokens from the audio RVQ tokenizer that only supports speech, aligning with MIO-Instruct's original training to avoid mode collapse. The results demonstrate meaningful improvements: while the baseline MIO-Instruct achieved 24.8% accuracy on OmniBench, fine-tuning with the vanilla setting improved performance to 25.7%, and the voice-filtered approach further enhanced accuracy to 29.2%. When we leverage the full OminiInstruct to LoRAfine-tune MiniCPM-o-2.6, the model improve its performance from 40.5% on to 45.9% on OmniBench. These results indicate that even a small subset of OmniInstruct can effectively adapt existing multimodal language models to tri-modal scenarios. Notably, the superior performance of the voice-filtered approach suggests that aligning fine-tuning data distribution with a model's distribution can yield better results.

## 5.3 Textual Approximation on Images and Audios

As the absence of strong OLM baselines, we further introduce the text alternatives of images ($I'$) and audios ($A'$) to embrace more dual-modal MLLMs to analyze current research progress.

**Performance Changes of Open OLMs.** We select the UnifiedIO-2 series to conduct the textual approximation experiments due to their relatively robust performances in the vanilla evaluation setting suggested in Table 3. Compared with the vanilla setting, all three UnifiedIO-2 models show performance gains, averagely at 6.42%, in the audio replacement setting and average performance drops in the replaced-image (1.87%) and both-repaced settings (0.12%). This indicates the shortcoming of existing OLMs on modeling the audio on the one hand, and the potential noise in the generated image captions compared to the human-written audio transcripts on the other hand.

**Performances of Dual-modal MLLMs.** In the setting of using text as the alternatives of audios and images, the VLMs show generally better results than ALMs (Table 6 vs Table 7) even compared with open-source model with similar model size[7]. This could be caused by : 1) more available research resources have been put in VLMs to develop datasets and cross-modality alignment architectures, leading to higher instruction following rate and accuracy compared

Table 6: Results on Textual Audio Approximation Experiments. All the audios are represented in text transcript. The results are divided into groups of vision-language models and omni-models. We use the text transcript to approximate the audios in this setting. Boldface shows the best model performance, and underline shows the best open-source model.

| Input Context | Image & Audio Transcript | Audio Transcript | Image |
|---|---|---|---|
| InternVL-2-2B | 42.29% | 27.32% | 28.11% |
| InternVL-2-8B | 50.79% | 33.63% | 33.36% |
| InternVL-2-26B | 51.75% | 31.87% | 33.89% |
| InternVL-2-40B | 54.29% | 31.96% | 34.76% |
| Qwen2-VL-Chat-2B | 42.47% | 31.44% | 38.09% |
| Qwen2-VL-Chat-7B | 48.60% | 32.05% | 36.87% |
| Deepseek-VL-Chat-7B | 39.67% | 29.51% | 26.27% |
| Idefics2-8B | 45.10% | 32.31% | 34.41% |
| Mantis-Idefics-8B | 46.15% | 36.43% | 32.57% |
| LLaVA-OneVision-0.5B | 38.00% | 31.79% | 31.17% |
| LLaVA-OneVision-7B | 47.02% | 31.70% | 29.68% |
| Cambrian-8B | 42.12% | 31.35% | 32.22% |
| Cambrian-13B | 45.01% | 31.96% | 33.98% |
| Cambrian-34B | 46.76% | 30.12% | 33.01% |
| XComposer2-4KHD (7B) | 43.96% | 29.25% | 30.65% |
| GPT4-o (0513) | 57.62% | 45.71% | 42.21% |
| GPT4-o (0806) | 51.14% | 47.55% | 31.44% |
| GPT4-o-mini | 49.04% | 39.23% | 34.06% |
| Gemini-1.5-Pro | 44.40% | 22.50% | 26.09% |
| Reka-core-20240501 | 46.58% | 34.59% | 30.65% |
| Claude-3.5-Sonnet | **59.37%** | 33.54% | **43.08%** |
| GPT-4V-Preview | 38.18% | 41.24% | 25.57% |
| GPT-4V-0409 | 33.36% | **45.80%** | 32.75% |
| UnifiedIO2-large (1.1B) | 34.33% | 31.96% | 29.07% |
| UnifiedIO2-xlarge (3.2B) | 43.17% | 34.50% | 34.76% |
| UnifiedIO2-xxlarge (6.8B) | 40.81% | 29.77% | 33.45% |

to ALMs; 2) the audio data are naturally harder (and hence more expensive) to annotate; and 3) audio typically has longer sequence tokens and requires more computational resource compared to text and image, making it harder to scale up. If $I'$ and $A'$ have the information loss ratio when converted from $I$ and $A$, it seems to be easier for the researchers to train the future omni-language models from exisiting VLMs rather than ALMs. Besides, we can observe Claude-3.5 and GPT-4o are generally the best two VLMs, significantly better compared to open-source VLMs. Qwen2- audio and Gemini are the two best ALMs in speech, Qwen2- audio is the best in sound, and Audio-SALMONN is music. Moreover, we can see significant differences in different types of audio, i.e., LTU and audio-flamingo are worse for music compared to speech, while Qwen-audio, which includes music on pre-training, provides better results on music compared to speech. And MusiLingo only uses music for pre-training performs worse in speech and audio.

---

[7]The results of pure text $(I', A')$ setting are placed at Table 8, Appendix A.

Table 7: Results on Textual Image Approximation Experiments. All the images are represented in text caption. The results are divided into groups of audio-language models and omni-models.

| Accuracy ↑ | All Audio Types | | | Speech | Sound Event | Music |
|---|---|---|---|---|---|---|
| **Input Context** | Image Caption & Audio | Audio | Image Caption | Image Caption & Audio | | |
| LTU (7B) | 23.29% | 23.91% | 23.12% | 25.42% | 20.00% | 16.04% |
| Mu-LLaMA (7B) | 1.58% | 1.84% | 1.84% | 1.56% | 1.13% | 2.83% |
| MusiLingo-long-v1 | 13.66% | 11.03% | 9.02% | 11.93% | 13.96% | 25.47% |
| Audio-SALMONN (13B) | 34.76% | 32.66% | 33.36% | 34.50% | 29.43% | **50.00%** |
| Qwen-Audio-Chat (7B) | 17.51% | 16.64% | 18.39% | 14.66% | 22.64% | 25.47% |
| Qwen2-Audio-7B-Instruct | **40.72%** | **35.20%** | **35.29%** | **40.60%** | **41.89%** | 38.68% |
| Audio-Flamingo (1.3B) | 24.78% | 23.82% | 24.78% | 26.98% | 21.51% | 16.98% |
| Gemini-1.5-Pro | 38.62% | 28.02% | 21.02% | 39.82% | 33.96% | 41.51% |
| Reka-core-20240501 | 29.42% | 23.12% | 26.27% | 28.53% | 29.43% | 35.85% |
| UnifiedIO2-large (1.1B) | 29.16% | 29.07% | 29.33% | 28.40% | 32.45% | 26.42% |
| UnifiedIO2-xlarge (3.2B) | 32.22% | 31.17% | 30.21% | 32.43% | 32.45% | 30.19% |
| UnifiedIO2-xxlarge (6.8B) | 32.05% | 32.49% | 27.15% | 31.13% | 38.87% | 21.70% |

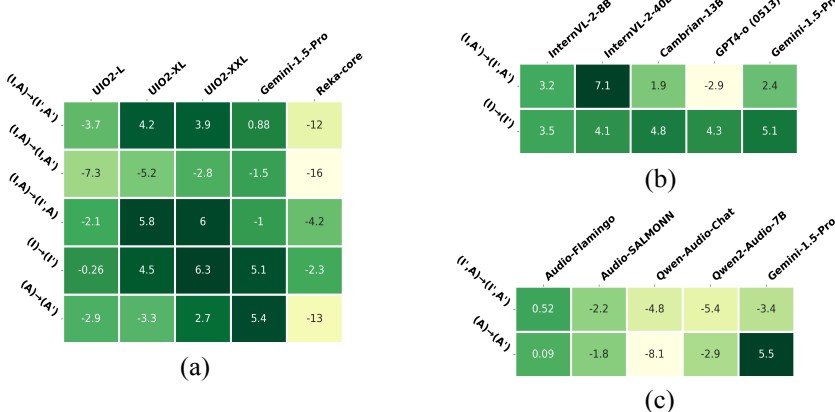

Figure 4: The Performance Changes Brought By Textual Alternatives. The numbers in the cell suggest the accuracy change. *(a)* includes the UnifiedIO2 OLMs and the proprietary models supporting tri-modal inputs. *(b)* and *(c)* consists of VLMs and ALMs grouped with Gemini-1.5-Pro for comparison.

**Pure Textual Evaluation.** The performance gaps brought by the replaced textual image and audio descriptions are in revealed in Figure 4. Notably, the majority of models demonstrate improved accuracy when processing textual representations of multimodal data compared to their performance on either image captions or audio transcripts alone. This suggests that these models show stronger reasoning capability when equipped with information from multiple textual sources rather than handling raw multimodal inputs. For instance, Qwen2-Audio-7B-Instruct shows a significant jump in accuracy from 39.05% (audio transcript only) and 39.67% (image caption only) to 47.02% when given both textual representations. But the gains of best proprietary models GPT4-o (0513) andClaude-3.5-Sonnet exhibit is not significant, though former achieved an impressive 60.60% in such setting.

# 6   Conclusion and Future Study

The proposed novel multimodal benchmark, OmniBench, reveals that current open-source multimodal large language models struggle with simultaneous processing of visual, acoustic, and textual inputs. We observed a general bias towards speech audio and superior performance of vision-language models over audio-language models when using textual approximations. These findings underscore the need for more appropriate architecture designs for multimodal integration, diverse datasets for training, and techniques to reduce modality bias. OmniBench serves as a crucial tool for guiding advancements in multimodal language models, driving progress towards more advanced and versatile models towards human-like multimodal understanding and reasoning.

## Impact Statement

This paper introduces OmniBench, a crucial step towards developing truly multimodal AI. By rigorously evaluating models on their ability to integrate visual, acoustic, and textual information, OmniBench exposes critical limitations in current approaches and highlights the need for dedicated research in tri-modal reasoning. This benchmark has significant potential societal impact, as robust OLMs could revolutionize fields like healthcare (improved diagnostics), accessibility (enhanced assistive technologies), and human-computer interaction (more natural interfaces). Furthermore, the accompanying OmniInstruct dataset provides a valuable resource for training and fine-tuning these models. While the development of OLMs raises ethical considerations regarding bias and fairness, OmniBench's focus on comprehensive evaluation enables researchers to identify and address these issues proactively. By fostering transparency and reproducibility, this work paves the way for responsible development and deployment of powerful multimodal AI systems that benefit society.

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

# A   More Experiment Results

Table 8: Results on Pure Textual Approximation for *Both Image and Audio*. All the images and audios are represented in texts. The results at the second and third column are taken from the corresponding models in Table 6 and Table 7.

| Input Context | Image Caption & Audio Transcript | Audio Transcript | Image Caption |
|---|---|---|---|
| LTU (7B) | 22.68% | 24.17% | 23.12% |
| Mu-LLaMA (7B) | 2.28% | 6.57% | 1.84% |
| MusiLIngo-long-v1 (7B) | 11.03% | 10.51% | 9.02% |
| Audio-SALMONN-13B | 36.95% | 34.41% | 33.36% |
| Qwen-Audio-Chat | 22.33% | 24.69% | 18.39% |
| Qwen2-Audio-7B-Instruct | 46.15% | 38.09% | 35.29% |
| Audio-Flamingo (1.3B) | 24.26% | 23.73% | 24.78% |
| InternVL-2-8B | 47.55% | 33.63% | 29.86% |
| InternVL-2-40B | 47.20% | 31.96% | 30.65% |
| Cambrian-13B | 43.08% | 31.96% | 29.16% |
| GPT4-o (0513) | 60.51% | 45.71% | 37.92% |
| GPT4-o (0806) | 53.77% | 47.55% | 29.51% |
| GPT4-o-mini | 51.05% | 49.04% | 32.84% |
| Gemini-1.5-Pro | 42.03% | 22.50% | 21.02% |
| Claude-3.5-Sonnet | 56.83% | 33.54% | 39.05% |
| Reka-core-20240501 | 42.23% | 36.33% | 32.94% |
| GPT-4V-Preview | 33.27% | 41.24% | 20.32% |
| GPT-4V-0409 | 29.95% | 45.80% | 20.84% |
| UnifiedIO2-large (1.1B) | 30.74% | 31.96% | 29.33% |
| UnifiedIO2-xlarge (3.2B) | 33.80% | 34.50% | 30.21% |
| UnifiedIO2-xxlarge (6.8B) | 34.15% | 29.77% | 27.15% |

# B   Dataset Development

## B.1   Statistics for OmniInstruct Dataset

Table 9: The Statistics of Data Filtering in OmniInstruct. The table shows the number changes of question-answer pairs before and after filtering from each of the data sources.

| Source | Original Train | Original Valid | Remained Train | Remained Valid |
|---|---|---|---|---|
| AVQA | 40,182 | 16,798 | 4,491 (11.2%) | 1,911 (11.4%) |
| Music-AVQA2.0 | 42,470 | 0 | 11 (0.03%) | 0 |
| MSRVTT-QA | 140,554 | 11,143 | 80,078 (57.0%) | 9,614 (86.2%) |
| Total | 233,206 | 27,941 | **84,580** | **11,525** |

As demonstrate in Table 9, most of the samples in the dataset are in low quality and therefore abandoned, and only 96k of samples remain for Omni-modality SFT training. This is reasonable because most of the questions are generated from templates, and the image may not sampled from the most relevant part of the questions and, therefore hot high in quality.

## B.2   Discussion of the Human Annotation Protocol

Question creation involved 16 annotators and 5 quality reviewers, covering music, linguistics, phonetics, cryptography, and math. During verification, annotators were required to document the modality-specific rationale for each question. Verifiers explicitly assessed this rationale for consistency. This supports our claim that questions are difficult, but not ill-defined. The final human test please refer to Appendix E.

## B.3   Diversity of Music Audios of OmniBenchmark

The music subset of our benchmark reflects a rich diversity of musical traditions, spanning a wide range of genres, styles, and cultural contexts. It encompasses Western classical symphonies, jazz chamber music, and avant-garde compositions alongside popular music from China, England, and

France. Traditional forms like Kunqu opera and modern experimental pieces are represented, as well as instrumental music from regions such as India, the Arab world, Africa, and Japan. The benchmark also includes famous film soundtracks with various thematic elements and Asian folk oral traditions, such as chanting, drumming, and Humai. This eclectic collection, enriched by unique instances like famous concert spoofs and iconic YouTube parodies, ensures that each question offers a distinct challenge, showcasing the nuanced intricacies and breadth of global music heritage.

## B.4    The Discussion of Video Data

While video data naturally encodes richer temporal dynamics, our current focus was to study targeted cross-modal reasoning using static images and audio. This choice enables finer control over modality composition and ambiguity—images can be selectively edited, cropped, or paired with audio to emphasize specific reasoning tasks (e.g., conflicting cues or missing information). At the time of benchmark development, only GPT-4o and Gemini offered reliable video+audio capabilities, whereas most open-source video models lacked support for audio streams, making it difficult for the current version of benchmark to include video-audio reasoning. Moreover, we believe that pruning the complexity of the temporal vision information could be a clearer experimental setting to analyze whether an OLM can better understand and reason based on the cross vision-audio information.

## B.5    Prompt for Quality Control on OmniInstruct Dataset

```
Initial Q&A: {question and answer}

The given Q&A is originally designed to answer based on the complementary context
built from an audio and an image together. Please evaluate whether the provided
Q&A is a bad/flawed sample due to one of the following reasons:

1. The answer could be inferred solely from the given image without the assistance
of audio;

2. The Q&A is not relevant to the image;

3. The Q&A is logically inconsistent.

After your evaluation, respond with 'Yes' if the Q&A is a flawed sample should be
removed, else response with 'No'.
```

Figure 5: The Prompt for OmniInstruct Dataset Filtering.

## B.6    Audio Transcription and Caption

We propose image caption as well as audio transcription and caption results to test ALMs and VLMs. For speech and song, we transcribe the speech text or lyrics. For music and other audio events that lacks natural language transcripts (e.g., instrumental music), we provide expert-written annotations capturing salient audio features such as instrumentation, emotion, dynamics, texture, genre, and production characteristics. These annotations are paired with the image and text to form the necessary context to answer the question. All music-based questions in OmniBench were manually vetted to ensure answerability using image and audio transcriptions alone.

We did not experiment with spectrogram-based embeddings for VLMs, though this maybe a promising direction. However, several technical limitations motivated our choice to use expert-written audio descriptions. On the one hand, spectrogram type and resolution vary by task: Speaker ID often requires 75–120 ms STFT windows; ASR uses 20–35 ms; pitch detection prefers CQT. A single universal spectrogram representation does not exist. On the other hand, interpretability: Expert-written audio summaries (e.g., instrumentation, emotion, genre, acoustics) paired with image and text offer a semantically compact, modality-aligned representation accessible to current vision-language models. Nevertheless, we highlight this as a direction for extending VLM compatibility with audio tasks.

## C   Data Annotation Platform

We conducted annotation correction and quality inspection using a professional annotation platform that supports structured editing, version control, and multi-stage review. A Snapshot of the platform interface is shown in Figure 6.

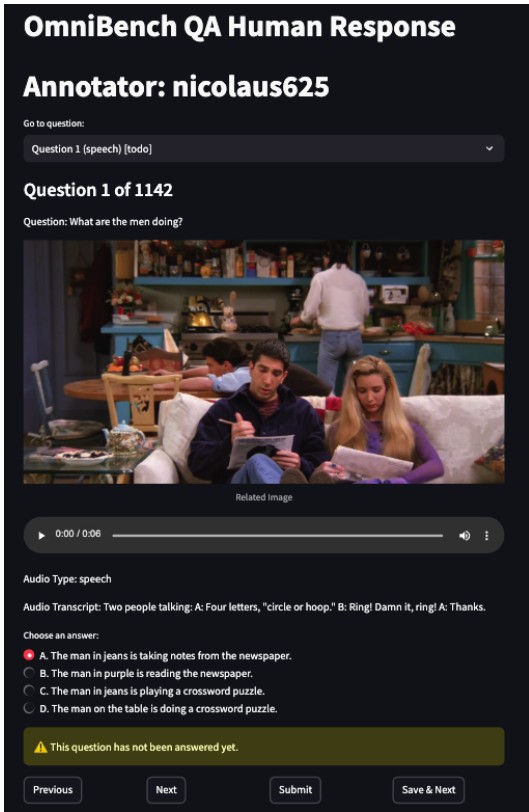

Figure 6: A Snapshot of the data annotation platform for human evaluation.

## D   Statistic Significance of the Findings

In this paper, we provide about 30 claims on the discussion in section 5. In order to discover whether the observation is due to the bias of a small test set, We utilize multiple types of hypothesis testing for these experimental results.

### D.1   Inference if Two Models Provide Significantly Different Results under the Same Small Subset

The method is inspired by [Guyon et al., 1998]. Suppose all the data samples $\{x_i\}_{i=1}^n$ are i.id, where $n$ is the number of data samples. For each specific model $M_j$, the performance of the $j^{th}$ model on the $i^{th}$ data is $M_j(x_i)$ which is equal to 0 if the model predicts the wrong choice and 1 otherwise. Due to the i.id. assumption of the test set, the $\{M_j(x_i)\}_{i=1}^n$ is a random variable sequence (r.v.s.) with binomial distribution with mean $p_i$ and variance $\sigma_i^2 = p_i(1 - p_i)$.

Suppose the accuracy $p_i$ of a given model is estimated by computing the average correct rate $\hat{p}_i$ over a finite number $n$ of test examples or patterns. When $n$ is small, $\hat{p_1} > \hat{p_2}$ may hold even when $p_1 < p_2$.

According to the normal law, the whited random variable

$$Z := \frac{p - \hat{p}}{\sigma\sqrt{n}}$$

obeys the standardized Normal law (with mean 0 and variance 1). Define the real number $Z_\alpha$ to be the value such that the probability $\mathbb{P}\left(x > Z_\alpha\right) = \alpha$. Then, according to Chebychev's inequality,

$$\mathbb{P}\left(p - \hat{p} \leq \frac{\sigma z_{alpha}}{\sqrt{n}}\right) \leq \alpha$$

.

We are then calculating the number of test examples $n$ that is needed to guarantee a certain margin of error $\epsilon_i$, e.g., $\epsilon(n, \alpha) = \frac{z_\alpha \sigma}{\sqrt{n}}$ for the Normal law). Denote $\beta := \frac{\epsilon_i}{p_i}$. Then

$$p - \hat{p} \leq z_\alpha \sqrt{\frac{p(1-p)}{n}}$$

hold with the probability $1 - \alpha$.

Therefore,

$$n \leq \left(\frac{z_\alpha}{\beta}\right)^2 \frac{1-p}{p}$$

holds with the probability $1 - \alpha$.

According to the cdf of standard normal distribution,

$$n\frac{\beta^2 p_1}{1 - p_1} = \frac{n(p_2 - p_1)^2}{p_1(1 - p_1)} \leq 2.72$$

implies significance 0.05, and the value $\leq 5.29$ implies p-value less than 0.01.

SImilarly, with Chernoff bound [Chernoff, 1952], $p - \hat{p} \leq \sqrt{\frac{-2p(\ln \alpha)}{n}}$ hold with the probability $1 - \alpha$. And therefore, $n = \frac{-2\ln \alpha}{\beta^2 p}$. The alternative estimation of p-value is $\alpha = \exp\left(\frac{np\beta^2}{2}\right)$

Therefore, we have the following claim with significance value:

- In Table 3
  - Claim (1): Scaling up may not contribute to the overall performance. With audio and image as input, UnifiedIO2-xlarge's performance 38.00% is better than the scaled-up UnifiedIO2-xxlarge's (33.98%) version on the 1142 samples holds, with a p-value is less than 0.01.
  - Claim (2): Performance f Gemini-1.5-Pro with image and audio input (42.91%) is better than single modality input (e.g. 27.93% for image only) on the 1142 samples holds with p-value 1.19e-20.
- Table 5
  - Claim (3): Scaling up of UnifiedIO can contribute to Count & Quantity tasks, though only 15 samples are selected. UnifiedIO-large (6.67%) is worse than UnifiedIO-xlarge (20%) holds with a p-value less than 0.05, and UnifiedIO-xlarge (20%) is worse than UnifiedIO-xxlarge (46.67%) holds with a p-value less than 0.01.
- In Table 4 & Table 6
  - - Claim (5): The performance of Claude-3.5 (59.37%) and GPT-4o (57.62%) in Table 4 is better than open-source VLM (less or equal to 54.29%). We can say Claude-3.5 is better than InternVL-2-40B (54.29%) with a p-value less than 0.05, and GPT-4o is better than the second-best open-source VLM InternVL-2-26B (51.75%) with p-value less than 0.01. But we can not say GPT-4o (57.62%) is better than InternVL-2-40B (54.29%).
  - Claim (6): **In speech subset (771 samples)** demonstrated in Table 4 with audio and image as input, Gemini-1.5-Pro (42.67%) does not surpass UnifiedIO2-xlarge (3.2B) (39.56%) significantly, but it surpass all other models like the second-best UnifiedIO2-xxlarge (6.8B) (34.24%) with p-value less than 0.01
  - Claim (7): **In general audio subset (265 samples)** demonstrated in Table 4 with audio and image as input, Gemini-1.5-Pro (42.26%) is better than all the others besides UnifiedIO2, such as Video-SALMONN (13B) (31.70%) with p-value less than 0.01

- Claim (8): **In the music subset (106 samples)** demonstrated in Table 4 with audio and image as input, Video-SALMONN (13B) (56.6%) is better than all the other models such as the best one Gemini-1.5-Pro (46.23%) with p-value less than 0.05.

- Compared with Table 7
  - Claim (9): **In the speech subset (771 samples)** demonstrated in Table 7 with audio and image caption as input, Qwen2-audio (40.60%) and Gemini (39.82%) are better than all other ALMs. E.g. Gemini is better than the best of the rest Audio-SALMONN (13B) (34.5%) with p-value less than 0.01
  - Claim (10): **In general audio subset (265 samples )** demonstrated in Table 7 with audio and image caption as input, Qwen2-audio (41.89%) is better than every model besides UnifiedIO2-xxlarge (38.87%), e.g. better than Gemini-1.5-Pro (33.96%) with p-value less than 0.05.
  - Claim (11): **In music subset (106 samples)** demonstrated in Table 7 with audio and image caption as input, Audio-SALMONN (50%) is better than all the others besides Gemini-1.5-Pro (41.5%). For example, Audio-SALMONN is better than the best of the rest Qwen2-Audio-7B-Instruct (38.68%) with a p-value less than 0.05.
  - Claim (12): In Table 7, with audio transcription and image caption as inputs, Qwen2-Audio-7B-Instruct (47.02%) is better than with only audio transcription (39.05%) or only image caption (39.67%) with a p-value less than 0.01
  - Claim (13): Replace the input of Qwen2-Audio-7B-Instruct from image caption and audio in Table 7 (40.72%) to caption and audio transcription in Table 6 (47.02%), the performance increases significantly with a p-value less than 0.01
  - Claim (14): Replace the input of Claude-3.5-Sonnet from image and audio transcription in Table 4 (59.37%) to image caption and transcription in Table 6 (56.83%) has no significant changes. And GPT4-o (0513) performances in Table 4 (57.62%) and Table 6 (60.51%) has no significant changes either.
  - Claim (15): GPT4-o (0513) performance in Table 6 (60.51%) has no significant difference with human (63.19%)

## D.2 Comparasion between Two Different Setting with the Same Samples

We use student-paired t-tests to compare two different experimental settings on the same testset.

Recall the performance of UnifiedIO2 in different experimental settings are as follows:

- performance with image and audio (25.94%, 39.56%, 34.24% in Table 2)
- performance with image caption & audio (29.16%, 32.22%, 32.05% in Table 7)
- performance with image & audio traiscripton (34.33%, 43.17%, 40.81% in Table 4)
- performance with pure text input (30.74%, 33.80%, 34.15% in Table 6)

Then the t-test results are as follows:

- between Table 2 and Table 4 with audio transcription has a p-value of 0.0471, showing the model has room for improvement in audio understanding capability compared with audio transcription ground truth.
- Claim (17): There is no significant difference between Table 3 and Table 6 (p-value 0.562) or Table 7 (p-value 0.919), showing the capability of OLM on image understanding is similar to the SOTA LLM generated image caption.

## D.3 Comparasion on the Same Setting with Two Different Subset Samples

We utilize Wilcoxon rank sum test (i.e. Mann–Whitney U test) to compare the model performance on two different subsets with different but independent samples.

- In Table 4 (top):
  - Claim (18): The performance of salmonn on the music subset (56.60%, 106 samples) is better than speech (34.11%, 771 samples) with p-value 6.84e-6; and better than sound events (31.70%, 265 samples) with p-value 8.98e-6.

- Claim (19): The performance of UnifiedIO2-xlarge (3.2B) music (29.25%) may not be worse than sound (36.98%) with p-value 0.158; but may be worse than speech (39.56%) with p-value 0.0407.
- Claim (20): The performance of Reka-core-20240501 on sound (26.04%) may not be worse than music (33.02%) with p-value 0.177; and may not be worse than speech (31.52%) with p-value 0.093.

- Table 4 (bottom)

  - Claim (21): the performance of Gemini-1.5-Pro on Story Description (30.87%, 230 samples) is worse than all the other 7 class with p-value 5.38e-5; Plot Inference (32.91%, 237 samples) worse than others with p-value 6.69e-4; Object Identification & Description (62.56%, 211 samples) is better than others with p-value 9.46e-11. This statement holds for many other models, but not for AnyGPT.
  - Claim (22): Gemini-1.5-Pro on Count & Quantity (13.33%, 15 samples) is worse than all the other 7 classes with a p-value of 0.0209. Reka-core-20240501 on Count & Quantity (6.67%) is worse than the rest with a p-value of 0.0445. Video-SALMONN on Count & Quantity (6.67%) is worse than the rest with a p-value of 0.0239. But not for AnyGPT.

- In Table 7

  - claim (23): The performance of LTU on music (16.04%) is worse than speech (25.42%) with a p-value of 0.0348, but may not be worse than sound (20.00%) with a p-value of 0.379.
  - claim (24): The performance of audio-flamingo on music (16.98%) is worse than speech (26.98%) with a p-value of 0.0275, but may not be worse than sound (21.51%) with a p-value of 0.328.
  - claim (25): The performance of Qwen-audio on music (25.47%) is better than speech (14.66%) with a p-value of 0.0044.
  - claim (26): The performance of MusiLingo on music (25.47%) is better than speech (11.93%) with a p-value of 1.37e-4, and better than sound (13.96%) with a p-value of 5.96e-3.

Given that we conducted 20-30 hypothesis tests, it is not surprising to observe at least one p-value below 0.05 due to the multiple comparisons, which could suggest a false positive in some cases. However, many of the tests involve the same models or subsets, meaning the p-values are not independent. As a result, the observed number of significant p-values ($< 0.04$, in fact, $< 0.01$ for many cases) is less indicative of false positives than it would be under the assumption of independent tests. This reduces the likelihood of false positives in our findings.

# E   Human Evaluation

All the questions is reviewed by evaluators (denoted as "human evaluator" or "human expert" in the main text) with X-minute answer time limit (presented in the draft) and another set of experts with more time permit (2-10 minutes), deriving a higher score of 74.03%. This includes the comparison between humans with strict and loose time limits to acquire responses to estimate difficulty vs. ambiguity.

Human performance is lower than expected due to several factors: (1) Annotation was conducted under time constraints, limiting thorough reasoning. (2) Annotators may lack relevant background knowledge—some were non-native English/French speakers, potentially affecting speech or lyric comprehension. (3) Annotators held only associate degrees, which may impact annotation quality in complex reasoning tasks such as music expertise.

# F   Limitation

Despite the strengths of OmniBench, several limitations remain. First, due to the high cost of annotation—each sample requiring approximately 30 minutes to design and 10 minutes to verify by human annotators—this version of the benchmark includes only 1,000 manually curated examples.

We plan to expand the dataset in future iterations. Second, the image captions used for text-alternative settings were generated by large language models rather than human annotators. While this approach improves scalability, it leads to suboptimal performance on tasks requiring precise visual text recognition (e.g., OCR-related questions), and the caption quality may not consistently surpass human-generated descriptions. Lastly, the current version of OmniBench focuses on static images and does not yet include video inputs. Extending the benchmark to support temporal reasoning with video and audio remains an important direction for future work.

# G   Experiments Compute Resources

The inference experiments were conducted using A800 GPUs, with each model processing the full OmniBench dataset over a period ranging from 30 minutes to 3 hours, depending on the model size and complexity. The evaluations were conducted via public APIs, which introduced variability due to network latency. The supervised fine-tuning on OmniInstruct dataset takes 5 minutes with LoRA using eight H800 GPUs for each epoch.

# H   Ethical Statement

**Human annotation and fair wages**: All annotators involved in data creation were either legally employed research assistants with music professionals or students supported by formal scholarships who are all co-authors of the paper, and were compensated in accordance with local minimum wage regulations. This aligns with NeurIPS requirements for fair compensation of human participants.

**Data privacy and consent**: All audio clips in MMAR were extracted from publicly accessible, user-uploaded videos on platforms like YouTube. Each clip is shorter than 30 seconds—shorter than preview segments on commercial platforms like Spotify—minimizing copyright and privacy risks. No personally identifiable or sensitive user information is included.

**Licensing and responsible use**: The dataset will be released under a CC-BY-NC license, explicitly limiting its use to non-commercial academic research, in accordance with NeurIPS guidelines for ethical dataset release and copyright respect.

**Diversity and representativeness**: OmniBench includes a balanced range of speech and vocal music data as discussed in subsection B.3, and the dataset covers multiple spoken and sung languages, including English, Chinese, Japanese, French, Italian, Spanish and Russian etc. This reflects an effort toward diversity and mitigates representational bias.

**Environmental and safety considerations**: The research poses no foreseeable risks related to safety, security, discrimination, surveillance, or environmental harm. The benchmark does not involve high-compute training or deployment of risky generative models.

# NeurIPS Paper Checklist

1. **Claims**

   Question: Do the main claims made in the abstract and introduction accurately reflect the paper's contributions and scope?

   Answer: [Yes]

   Justification: Yes, the abstract and introduction accurately reflect the paper's contributions and scope. The introduction of OmniBench and OmniInstruct aligns with the stated goal of evaluating and improving tri-modal reasoning in MLLMs. The main findings—performance limitations of open-source models, fragility of proprietary models, and human-model performance gaps—are consistent with the claims made. Overall, the abstract provides a clear and faithful summary of the paper's objectives, methods, and insights.

   Guidelines:

   - The answer NA means that the abstract and introduction do not include the claims made in the paper.
   - The abstract and/or introduction should clearly state the claims made, including the contributions made in the paper and important assumptions and limitations. A No or NA answer to this question will not be perceived well by the reviewers.
   - The claims made should match theoretical and experimental results, and reflect how much the results can be expected to generalize to other settings.
   - It is fine to include aspirational goals as motivation as long as it is clear that these goals are not attained by the paper.

2. **Limitations**

   Question: Does the paper discuss the limitations of the work performed by the authors?

   Answer: [Yes]

   Justification: The limitation is discussed in Appendix F. These reflections demonstrate awareness of the benchmark's scope and encourage future improvement.

   Guidelines:

   - The answer NA means that the paper has no limitation while the answer No means that the paper has limitations, but those are not discussed in the paper.
   - The authors are encouraged to create a separate "Limitations" section in their paper.
   - The paper should point out any strong assumptions and how robust the results are to violations of these assumptions (e.g., independence assumptions, noiseless settings, model well-specification, asymptotic approximations only holding locally). The authors should reflect on how these assumptions might be violated in practice and what the implications would be.
   - The authors should reflect on the scope of the claims made, e.g., if the approach was only tested on a few datasets or with a few runs. In general, empirical results often depend on implicit assumptions, which should be articulated.
   - The authors should reflect on the factors that influence the performance of the approach. For example, a facial recognition algorithm may perform poorly when image resolution is low or images are taken in low lighting. Or a speech-to-text system might not be used reliably to provide closed captions for online lectures because it fails to handle technical jargon.
   - The authors should discuss the computational efficiency of the proposed algorithms and how they scale with dataset size.
   - If applicable, the authors should discuss possible limitations of their approach to address problems of privacy and fairness.
   - While the authors might fear that complete honesty about limitations might be used by reviewers as grounds for rejection, a worse outcome might be that reviewers discover limitations that aren't acknowledged in the paper. The authors should use their best judgment and recognize that individual actions in favor of transparency play an important role in developing norms that preserve the integrity of the community. Reviewers will be specifically instructed to not penalize honesty concerning limitations.

3. **Theory assumptions and proofs**

   Question: For each theoretical result, does the paper provide the full set of assumptions and a complete (and correct) proof?

   Answer: [NA]

   Justification: The paper does not present any theoretical results requiring formal assumptions or mathematical proofs. The only formulaic reference is the multiple methods on hypothesis testing, which is a standard statistical adjustment rather than a novel theoretical contribution. We provide the details calculation in the appendix.

   Guidelines:

   - The answer NA means that the paper does not include theoretical results.
   - All the theorems, formulas, and proofs in the paper should be numbered and cross-referenced.
   - All assumptions should be clearly stated or referenced in the statement of any theorems.
   - The proofs can either appear in the main paper or the supplemental material, but if they appear in the supplemental material, the authors are encouraged to provide a short proof sketch to provide intuition.
   - Inversely, any informal proof provided in the core of the paper should be complemented by formal proofs provided in appendix or supplemental material.
   - Theorems and Lemmas that the proof relies upon should be properly referenced.

4. **Experimental result reproducibility**

   Question: Does the paper fully disclose all the information needed to reproduce the main experimental results of the paper to the extent that it affects the main claims and/or conclusions of the paper (regardless of whether the code and data are provided or not)?

   Answer: [Yes]

   Justification: The paper provides a comprehensive and transparent account of all components necessary to reproduce its main experimental results. The OmniBench benchmark and OmniInstruct dataset, including both JSON annotations and audio files, is publicly hosted on Hugging Face, and the evaluation script is available on GitHub. Detailed descriptions are provided for all evaluated models that are either open-sourced or have APIS available, including citations and categorisation across different input modalities. The paper also clearly explains the evaluation procedure, including how image captions are generated and used as inputs to non-visual models. This level of detail ensures reproducibility of both the experimental setup and the main benchmarking results.

   Guidelines:

   - The answer NA means that the paper does not include experiments.
   - If the paper includes experiments, a No answer to this question will not be perceived well by the reviewers: Making the paper reproducible is important, regardless of whether the code and data are provided or not.
   - If the contribution is a dataset and/or model, the authors should describe the steps taken to make their results reproducible or verifiable.
   - Depending on the contribution, reproducibility can be accomplished in various ways. For example, if the contribution is a novel architecture, describing the architecture fully might suffice, or if the contribution is a specific model and empirical evaluation, it may be necessary to either make it possible for others to replicate the model with the same dataset, or provide access to the model. In general. releasing code and data is often one good way to accomplish this, but reproducibility can also be provided via detailed instructions for how to replicate the results, access to a hosted model (e.g., in the case of a large language model), releasing of a model checkpoint, or other means that are appropriate to the research performed.
   - While NeurIPS does not require releasing code, the conference does require all submissions to provide some reasonable avenue for reproducibility, which may depend on the nature of the contribution. For example
     (a) If the contribution is primarily a new algorithm, the paper should make it clear how to reproduce that algorithm.

(b) If the contribution is primarily a new model architecture, the paper should describe the architecture clearly and fully.

(c) If the contribution is a new model (e.g., a large language model), then there should either be a way to access this model for reproducing the results or a way to reproduce the model (e.g., with an open-source dataset or instructions for how to construct the dataset).

(d) We recognize that reproducibility may be tricky in some cases, in which case authors are welcome to describe the particular way they provide for reproducibility. In the case of closed-source models, it may be that access to the model is limited in some way (e.g., to registered users), but it should be possible for other researchers to have some path to reproducing or verifying the results.

5. **Open access to data and code**

Question: Does the paper provide open access to the data and code, with sufficient instructions to faithfully reproduce the main experimental results, as described in supplemental material?

Answer: [Yes]

Justification: The paper provides open access to both the dataset and evaluation code. The OmniBench benchmark and OmniInstruct dataset, including audio files and JSON annotations, is released on Hugging Face, while the evaluation script and usage instructions are hosted on GitHub.

Guidelines:

- The answer NA means that paper does not include experiments requiring code.
- Please see the NeurIPS code and data submission guidelines (`https://nips.cc/public/guides/CodeSubmissionPolicy`) for more details.
- While we encourage the release of code and data, we understand that this might not be possible, so "No" is an acceptable answer. Papers cannot be rejected simply for not including code, unless this is central to the contribution (e.g., for a new open-source benchmark).
- The instructions should contain the exact command and environment needed to run to reproduce the results. See the NeurIPS code and data submission guidelines (`https://nips.cc/public/guides/CodeSubmissionPolicy`) for more details.
- The authors should provide instructions on data access and preparation, including how to access the raw data, preprocessed data, intermediate data, and generated data, etc.
- The authors should provide scripts to reproduce all experimental results for the new proposed method and baselines. If only a subset of experiments are reproducible, they should state which ones are omitted from the script and why.
- At submission time, to preserve anonymity, the authors should release anonymized versions (if applicable).
- Providing as much information as possible in supplemental material (appended to the paper) is recommended, but including URLs to data and code is permitted.

6. **Experimental setting/details**

Question: Does the paper specify all the training and test details (e.g., data splits, hyperparameters, how they were chosen, type of optimizer, etc.) necessary to understand the results?

Answer: [Yes]

Justification: The paper specifies all relevant experimental details necessary to understand and reproduce the results. The dataset curation process is extensively documented, including how questions, answers, and image/audio transcription were created, validated, and quality-checked. The evaluation setting clearly defines the task formulation (multiple-choice), metric (classification accuracy), and model input structure (audio + question + choices), along with the curation of OmniInstruct dataset. For ALM and VLM, the paper describes how image/audio is first converted to captions before being fed into language models. Although the paper does not involve model training and therefore omits hyperparameter or optimizer details, it thoroughly covers all required test-time settings. Model categories, individual model sources, and evaluation procedures are well-documented.

Guidelines:

- The answer NA means that the paper does not include experiments.
- The experimental setting should be presented in the core of the paper to a level of detail that is necessary to appreciate the results and make sense of them.
- The full details can be provided either with the code, in appendix, or as supplemental material.

7. **Experiment statistical significance**

Question: Does the paper report error bars suitably and correctly defined or other appropriate information about the statistical significance of the experiments?

Answer: [Yes]

Justification: The paper provides a detailed description of three different types of hypothesis testing in Appendix D.

Guidelines:

- The answer NA means that the paper does not include experiments.
- The authors should answer "Yes" if the results are accompanied by error bars, confidence intervals, or statistical significance tests, at least for the experiments that support the main claims of the paper.
- The factors of variability that the error bars are capturing should be clearly stated (for example, train/test split, initialization, random drawing of some parameter, or overall run with given experimental conditions).
- The method for calculating the error bars should be explained (closed form formula, call to a library function, bootstrap, etc.)
- The assumptions made should be given (e.g., Normally distributed errors).
- It should be clear whether the error bar is the standard deviation or the standard error of the mean.
- It is OK to report 1-sigma error bars, but one should state it. The authors should preferably report a 2-sigma error bar than state that they have a 96% CI, if the hypothesis of Normality of errors is not verified.
- For asymmetric distributions, the authors should be careful not to show in tables or figures symmetric error bars that would yield results that are out of range (e.g. negative error rates).
- If error bars are reported in tables or plots, The authors should explain in the text how they were calculated and reference the corresponding figures or tables in the text.

8. **Experiments compute resources**

Question: For each experiment, does the paper provide sufficient information on the computer resources (type of compute workers, memory, time of execution) needed to reproduce the experiments?

Answer: [Yes]

Justification: Information discussed in Appendix G. While exact memory and storage configurations are not specified for each run, the provided information is sufficient to estimate the compute scale and replicate the experimental setup.

Guidelines:

- The answer NA means that the paper does not include experiments.
- The paper should indicate the type of compute workers CPU or GPU, internal cluster, or cloud provider, including relevant memory and storage.
- The paper should provide the amount of compute required for each of the individual experimental runs as well as estimate the total compute.
- The paper should disclose whether the full research project required more compute than the experiments reported in the paper (e.g., preliminary or failed experiments that didn't make it into the paper).

9. **Code of ethics**

Question: Does the research conducted in the paper conform, in every respect, with the NeurIPS Code of Ethics https://neurips.cc/public/EthicsGuidelines?

Answer: [Yes]

Justification: The research conducted in the paper conforms to the NeurIPS Code of Ethics in all respects, including ethical data sourcing, fair labor practices, legal compliance, privacy protection, and responsible dissemination in Appendix H By adhering to these ethical standards throughout data collection, annotation, and release, the work aligns with the spirit and letter of the NeurIPS Code of Ethics.

Guidelines:

- The answer NA means that the authors have not reviewed the NeurIPS Code of Ethics.
- If the authors answer No, they should explain the special circumstances that require a deviation from the Code of Ethics.
- The authors should make sure to preserve anonymity (e.g., if there is a special consideration due to laws or regulations in their jurisdiction).

10. **Broader impacts**

Question: Does the paper discuss both potential positive societal impacts and negative societal impacts of the work performed?

Answer: [NA]

Justification: The paper presents a benchmark dataset for academic research and does not involve the development or deployment of models with direct real-world applications. It does not include personal data, sensitive content, or generative components, and all audio clips are sourced from publicly available material and limited to under 30 seconds. Given its foundational nature and restricted non-commercial license (CC-BY-NC), the work does not pose immediate societal impact risks.

Guidelines:

- The answer NA means that there is no societal impact of the work performed.
- If the authors answer NA or No, they should explain why their work has no societal impact or why the paper does not address societal impact.
- Examples of negative societal impacts include potential malicious or unintended uses (e.g., disinformation, generating fake profiles, surveillance), fairness considerations (e.g., deployment of technologies that could make decisions that unfairly impact specific groups), privacy considerations, and security considerations.
- The conference expects that many papers will be foundational research and not tied to particular applications, let alone deployments. However, if there is a direct path to any negative applications, the authors should point it out. For example, it is legitimate to point out that an improvement in the quality of generative models could be used to generate deepfakes for disinformation. On the other hand, it is not needed to point out that a generic algorithm for optimizing neural networks could enable people to train models that generate Deepfakes faster.
- The authors should consider possible harms that could arise when the technology is being used as intended and functioning correctly, harms that could arise when the technology is being used as intended but gives incorrect results, and harms following from (intentional or unintentional) misuse of the technology.
- If there are negative societal impacts, the authors could also discuss possible mitigation strategies (e.g., gated release of models, providing defenses in addition to attacks, mechanisms for monitoring misuse, mechanisms to monitor how a system learns from feedback over time, improving the efficiency and accessibility of ML).

11. **Safeguards**

Question: Does the paper describe safeguards that have been put in place for responsible release of data or models that have a high risk for misuse (e.g., pretrained language models, image generators, or scraped datasets)?

Answer: [NA]

Justification: The paper does not release any models or data with high risk for misuse. The OmniBench benchmark and OmniInstruct dataset contains short (<30s) audio clips sourced from publicly available videos, with no personal or sensitive content. It does not involve the release of generative models or systems that could be repurposed for harmful applications. As such, safeguards for high-risk assets are not necessary in this context.

Guidelines:

- The answer NA means that the paper poses no such risks.
- Released models that have a high risk for misuse or dual-use should be released with necessary safeguards to allow for controlled use of the model, for example by requiring that users adhere to usage guidelines or restrictions to access the model or implementing safety filters.
- Datasets that have been scraped from the Internet could pose safety risks. The authors should describe how they avoided releasing unsafe images.
- We recognize that providing effective safeguards is challenging, and many papers do not require this, but we encourage authors to take this into account and make a best faith effort.

12. **Licenses for existing assets**

Question: Are the creators or original owners of assets (e.g., code, data, models), used in the paper, properly credited and are the license and terms of use explicitly mentioned and properly respected?

Answer: [Yes]

Justification: All existing assets used in OmniBench, including pre-trained models and datasets, are properly cited with references to their original publications. The terms of use for each asset are respected, and our benchmark is released under the MIT license. Audio data is sourced from publicly available, user-uploaded internet content, and is used under fair-use considerations with additional restrictions (e.g., <30 seconds, CC-BY-NC license) to minimize copyright risks. The OmniInstruct dataset is based on AVQA (non-commercial license), MSRVTT-QA (MIT license), and MUSIC-AVQA-v2.0 (license), and therefore, we setup the OmniInstruct as cc-by-nc license as well.

Guidelines:

- The answer NA means that the paper does not use existing assets.
- The authors should cite the original paper that produced the code package or dataset.
- The authors should state which version of the asset is used and, if possible, include a URL.
- The name of the license (e.g., CC-BY 4.0) should be included for each asset.
- For scraped data from a particular source (e.g., website), the copyright and terms of service of that source should be provided.
- If assets are released, the license, copyright information, and terms of use in the package should be provided. For popular datasets, `paperswithcode.com/datasets` has curated licenses for some datasets. Their licensing guide can help determine the license of a dataset.
- For existing datasets that are re-packaged, both the original license and the license of the derived asset (if it has changed) should be provided.
- If this information is not available online, the authors are encouraged to reach out to the asset's creators.

13. **New assets**

Question: Are new assets introduced in the paper well documented and is the documentation provided alongside the assets?

Answer: [Yes]

Justification: The paper introduces a new benchmark (OmniBench) and a new dataset (OmniInstruct). all associated assets—including audio files, annotations, and evaluation scripts—are well documented in the README files of both the Hugging Face dataset repository and the GitHub codebase.

Guidelines:

- The answer NA means that the paper does not release new assets.
- Researchers should communicate the details of the dataset/code/model as part of their submissions via structured templates. This includes details about training, license, limitations, etc.
- The paper should discuss whether and how consent was obtained from people whose asset is used.
- At submission time, remember to anonymize your assets (if applicable). You can either create an anonymized URL or include an anonymized zip file.

14. **Crowdsourcing and research with human subjects**

    Question: For crowdsourcing experiments and research with human subjects, does the paper include the full text of instructions given to participants and screenshots, if applicable, as well as details about compensation (if any)?

    Answer: [Yes]

    Justification: The paper does not involve crowdsourcing or external human subject experiments. All annotations were performed by the authors themselves or employees from the same company as the correspondent authors, being paid much more than the minimum wage in the region of the data collector. The process of annotation and human evaluation is discussed in the paper. To ensure transparency, the paper includes screenshots of the annotation interface in Appendix C.

    Guidelines:

    - The answer NA means that the paper does not involve crowdsourcing nor research with human subjects.
    - Including this information in the supplemental material is fine, but if the main contribution of the paper involves human subjects, then as much detail as possible should be included in the main paper.
    - According to the NeurIPS Code of Ethics, workers involved in data collection, curation, or other labor should be paid at least the minimum wage in the country of the data collector.

15. **Institutional review board (IRB) approvals or equivalent for research with human subjects**

    Question: Does the paper describe potential risks incurred by study participants, whether such risks were disclosed to the subjects, and whether Institutional Review Board (IRB) approvals (or an equivalent approval/review based on the requirements of your country or institution) were obtained?

    Answer: [NA]

    Justification: All data collection and annotation were conducted by the authors and qualified collaborators or the employees of the company. This study has been reviewed and approved by the Human and Artefacts Research Ethics Committee of legal team at 01.ai, ensuring our research adheres to ethical guidelines in data usage, AI generation, and cultural representation. No direct interaction with participants occurred, and no personally identifiable information was collected.

    Guidelines:

    - The answer NA means that the paper does not involve crowdsourcing nor research with human subjects.
    - Depending on the country in which research is conducted, IRB approval (or equivalent) may be required for any human subjects research. If you obtained IRB approval, you should clearly state this in the paper.
    - We recognize that the procedures for this may vary significantly between institutions and locations, and we expect authors to adhere to the NeurIPS Code of Ethics and the guidelines for their institution.
    - For initial submissions, do not include any information that would break anonymity (if applicable), such as the institution conducting the review.

16. **Declaration of LLM usage**

    Question: Does the paper describe the usage of LLMs if it is an important, original, or non-standard component of the core methods in this research? Note that if the LLM is used only for writing, editing, or formatting purposes and does not impact the core methodology, scientific rigorousness, or originality of the research, declaration is not required.

    Answer: [Yes]

    Justification: LLMS were used to determine whether the multiple-choice options of each question are confusing enough. Besides, the image caption is generated by VLM to evaluate the performance of ALM. Moreover, VLM is used to generate and filter the data quality of the OmniInstruct dataset.

    Guidelines:

    - The answer NA means that the core method development in this research does not involve LLMs as any important, original, or non-standard components.
    - Please refer to our LLM policy (`https://neurips.cc/Conferences/2025/LLM`) for what should or should not be described.

