# OpenReview forum: "OmniBench: Towards The Future of Universal Omni-Language Models"
_NeurIPS.cc/2025/Datasets_and_Benchmarks_Track — NeurIPS 2025 Datasets and Benchmarks Track poster_

### Official Review · Reviewer_9aYp · 2025-06-30

**Rating:** 5
**Confidence:** 4

**Summary:**

This paper proposes the OmniBench, a benchmark evaluate the OLM's ability across visual, acoustic, and textual. Besides, the authors curate OmniInstruct to train OLMs to adapt to tri-modal contexts.

**Dataset Code Accessibility:**

Yes

**Ethical Considerations:**

No, there are no or only very minor ethics concerns

**Final Justification:**

The authors resolved all my concerns.

**Limitations Weaknesses:**

1. **Missing in-depth analysis of OmniBench.** The category in OmniBench can be divided into different categories, but the paper does not provide the data distribution of each category. Besides, the reason for introducing each type of question is not explained.
2. **The quality of OmniInstruct.** This paper uses InternVL-2-76B to filter three simple questions. However, it remains questionable whether filtering simple questions solely based on model performance is reliable. Besides, the effect of simple questions is not yet discussed.
3. This automated process evaluates each sample under three settings. For audios without transcript, such as music without lyrics, how to convert such audios for LLaVA-1.6-34B?
4. Typos: missed reference in Line 137.

**Strengths Contributions:**

1. This paper proposes the omnibench to evaluate the performance of OLMs, acting as a standard to quantify the model’s performance.
2. The authors develop the OmniInstruct to improve the model capability of tri-modal reasoning.

---

> ### Author Rebuttal · Authors · 2025-07-31
>
> **We thank the reviewer for the constructive feedback and for acknowledging the contributions of OmniBench and OmniInstruct. Below, we address each of the concerns in detail.**
>
> ### 1. In-depth analysis of OmniBench categories and data distribution
>
> OmniBench was designed with a structured taxonomy reflecting narrative and situational understanding, consisting of eight categories: *Current Action & Activity, Story Description with Causal, Plot Inference, Object Identification & Description, Contextual & Environmental, Identity & Relationship, Text & Symbols,* and *Count & Quantity*. This taxonomy stems from classical narrative elements—*who, what, when, where, why, how*—augmented with abstract reasoning (logical, symbolic, numerical). For clarity, we will include the category-wise sample distribution in the revised version (see the attached table), and expand the motivation for each category in Section 3.1 to better guide the reader through the design rationale.
>
> ### 2. Justification and effect of simple-question filtering in OmniInstruct
>
> We selected InternVL-2-76B—the strongest open-source VLM at the time of curation—as a proxy to assess whether a question could be unambiguously answered **without audio information**. This filtering is not based on model correctness, but rather whether the model determines the answer is **non-unique** without access to audio. Thus, the retained examples require true tri-modal reasoning.
>
> While this method is imperfect, our experiments demonstrate that removing such “simple” cases significantly improves downstream performance: MIO-Instruct baseline rises from 24.8% to 29.2%, and MiniCPM-o-2.6 with LoRA improves from 40.5% to 45.9% with full OmniInstruct. We will clarify this methodology in Section 4.
>
> ### 3. Handling of music audio without transcripts
>
> We thank the reviewer for pointing out this ambiguity. For music and other audio events that lacks natural language transcripts (e.g., instrumental music), we provide expert-written annotations capturing salient audio features such as *instrumentation, emotion, dynamics, texture, genre,* and *production characteristics*. These annotations are paired with the image and text to form the necessary context to answer the question. All music-based questions in OmniBench were manually vetted to ensure answerability using image and audio transcriptions alone. We will elaborate on this annotation strategy in Section 3.2 and add example questions involving music in Appendix B.
>
> ### 4. Missing reference on Line 137
>
> We thank the reviewer for flagging this typo. We will correct the missing citation in Line 137 and perform a thorough proofread for any remaining issues.
>
>
>
> | Statistic | Current Action & Activity | Story Description | Plot Inference | Object Identification & Description | Contextual & Environmental | Identity & Relationship | Text & Symbols | Count & Quantity | Overall |
> |---|---|---|---|---|---|---|---|---|---|
> | Quantity |  |  |  |  |  |  |  |  |  |
> | Total | 251 | 230 | 237 | 211 | 141 | 32 | 25 | 15 | 1142 |
> | Speech | 78 | 182 | 179 | 162 | 104 | 31 | 22 | 13 | 771 |
> | Sound Event | 147 | 27 | 37 | 28 | 25 | 1 | - | - | 265 |
> | Music | 26 | 21 | 21 | 21 | 12 | - | 3 | 2 | 106 |
> | Word Length |  |  |  |  |  |  |  |  |  |
> | Question | 4.68 | 5.75 | 7.47 | 7 | 6.85 | 6.22 | 7.32 | 8.72 | 6.25 |
> | Option | 8.85 | 7.77 | 8.92 | 6.47 | 5.68 | 10.38 | 11.22 | 6.6 | 8.81 |
> | Img. Rationale | 18.27 | 19.62 | 24.4 | 24.94 | 18.34 | 22.69 | 24.8 | 29.16 | 21.19 |
> | Audio Rationale | 23.11 | 20.5 | 24.4 | 20.97 | 18.27 | 24.92 | 23.1 | 53.84 | 22.9 |
> | Audio Content | 13.21 | 17.91 | 29.87 | 28.03 | 14.41 | 19.01 | 23.31 | 35.16 | 18.37 |
> | Multimodal Info. |  |  |  |  |  |  |  |  |  |
> | Img. Width | 1283.75 | 1291.6 | 2394.93 | 1430.03 | 1141.39 | 1395.53 | 1338.51 | 1787.36 | 1322.36 |
> | Img. Height | 842.32 | 776.11 | 2089.93 | 799.47 | 728.06 | 840.15 | 761.58 | 1168.04 | 818.64 |
> | Audio Len. (s) | 7.35 | 9.82 | 11.22 | 11.43 | 8.03 | 8.63 | 11.43 | 15.63 | 9.22 |

---

> > ### Comment · Reviewer_9aYp · 2025-08-06
> >
> > The authors have addressed all my concerns, I'll maintain my scores.

---

> ### Comment · Area_Chair_GKmb · 2025-08-05
>
> Please check if your concerns have been addressed by the authors.

---

### Official Review · Reviewer_f9w2 · 2025-06-30

**Rating:** 5
**Confidence:** 3

**Summary:**

OmniBench introduces a novel benchmark for evaluating multimodal large language models (MLLMs) on ​​tri-modal reasoning​​ (image, audio, and text). The benchmark features 1,142 human-annotated samples requiring integrated understanding across all three modalities.

**Additional Feedback:**

I am willing to increase my rating if the author could resolve my concerns.

**Dataset Code Accessibility:**

NA; not applicable to this submission (e.g., no new dataset, benchmark, code, or data provided)

**Ethical Comments:**

I am willing to increase my rating if the author could resolve my concerns.

**Ethical Considerations:**

No, there are no or only very minor ethics concerns

**Final Justification:**

The author addressed my concerns and I believe this paper could inspire the community.

**Limitations Weaknesses:**

**Questions for Authors:**

​​Bias in Music Tasks​​: Human experts significantly outperformed models on music-related queries (Table 3). Could this stem from dataset imbalances (e.g., genre diversity) or model architectural limitations?

​​Model-Centric vs. Benchmark-Centric Contribution​​: Beyond curating OmniBench/OmniInstruct, what specific architectural innovations do you propose for improving tri-modal fusion in future OLMs?

​​Temporal Reasoning​​: Why exclude video inputs? Would extending OmniBench to spatiotemporal tasks better reflect real-world multimodal scenarios?

​​Human Evaluation​​: Human annotators (musicians) achieved 63% accuracy, far higher than most models. Could this gap partly reflect task design (e.g., reliance on specialized knowledge)?

​​Beyond Text Approximation​​: For non-OLMs, did you explore alternative cross-modal alignment techniques (e.g., spectrogram embeddings for VLMs) instead of text captions?

**Strengths Contributions:**

1. OmniBench addresses a critical gap in multimodal research by introducing the first benchmark for ​​simultaneous tri-modal reasoning​​. Its rigorous annotation protocol (three-stage quality control) ensures high-quality, modality-interdependent samples.  The creation of ​​OmniInstruct​​ and the analysis of modality biases (e.g., speech vs. music performance) offer concrete pathways for future model improvement.

2. Comprehensive experiments test >20 models across multiple settings (full tri-modal, ablated modalities, text approximations), providing deep insights into current limitations.

3. This paper is well-written and presented.

---

> ### Author Rebuttal · Authors · 2025-07-31
>
> We sincerely thank the reviewer for the detailed and insightful feedback. We are encouraged by the recognition of OmniBench’s novelty, high-quality annotations, and diagnostic analyses. Below we address the reviewer’s questions point by point and appreciate the openness to raising the score.
>
>
> ### **1. Bias in Music Tasks**
>
> We agree that model underperformance on music-related questions (Table 3) is notable and multifactorial. While OmniBench was carefully curated to balance genre and linguistic diversity (e.g., avoiding overrepresentation of English pop), many audio-capable LLMs suffer from training-data bias. For instance, Qwen2-Audio leverages MTG-Jamendo for music tagging, yielding strong results there but poor generalization to other datasets like MTT (as shown in CMI-Bench [1]). In contrast to speech data, music audio is more restricted by copyright and expensive to annotate for supervised tasks.
>
> ### **2. Model-Centric vs. Benchmark-Centric Contribution**
>
> This paper focuses on benchmark construction and diagnostic evaluation, rather than architectural innovation. That said, our findings offer several directions for improving OLMs:
>
> - **Data-centric cross-modal alignment**: OmniBench’s strongest-performing model (Qwen-Omni) also leads in audio reasoning. Interestingly, it combines separate audio and vision instruction tuning, suggesting that fusing different bimodal corpora can generalize to tri-modal settings.
>
> - **Model-centric design**: Although most top-performing models (e.g., Video-SALMONN) still rely on modality-specific encoders, emerging models like **AnyGPT**—which use a universal tokenizer for multiple modalities—represent a more integrated architecture. We hypothesize that GPT-4o may follow this direction, and believe such designs will be necessary to scale OLMs to more complex understanding and generation tasks.
>
> We will highlight these implications more explicitly in the discussion section.
>
> ### **3. Temporal Reasoning and Exclusion of Video**
>
> While **video data naturally encodes richer temporal dynamics**, our current focus was to study *targeted cross-modal reasoning* using static images and audio. This choice enables **finer control over modality composition and ambiguity**—images can be selectively edited, cropped, or paired with audio to emphasize specific reasoning tasks (e.g., conflicting cues or missing information). At the time of benchmark development, **only GPT-4o and Gemini offered reliable video+audio capabilities**, whereas most open-source video models lacked support for audio streams, making it difficult for the current version of benchmark to include video-audio reasoning. Moreover, we believe that pruning the complexity of the temporal vision information could be a clearer experimental setting to analyze whether an OLM can better understand and reason based on the cross vision-audio information.
>
> The future work can include synchronized video-audio-text settings in future work.
>
> ### **4. Human Evaluation and Specialized Knowledge**
>
> You are right. The performance gap between models and human annotators underscores both the **difficulty and domain specificity** of certain tasks. Many music-related questions require nuanced understanding of music theory, instrumentation, or historical context—skills models typically lack unless explicitly trained. That said, not all questions depend on such knowledge: some involve **cross-modal inference**, such as recognizing mismatched emotional cues (e.g., upbeat music in a horror scene) or **statistical reasoning**, such as counting instrument entrances per bar. These tasks are accessible but still challenge current models’ multi-modal reasoning capabilities.
>
> ### **5. Beyond Text Approximation for Non-OLMs**
>
> We did not experiment with spectrogram-based embeddings for VLMs, and we agree this is a promising direction. However, several technical limitations motivated our choice to use **expert-written audio descriptions**:
>
> - **Spectrogram type and resolution vary by task**: Speaker ID often requires 75–120 ms STFT windows; ASR uses 20–35 ms; pitch detection prefers CQT. A single universal spectrogram representation does not exist.
>
> - **Interpretability**: Expert-written audio summaries (e.g., instrumentation, emotion, genre, acoustics) paired with image and text offer a semantically compact, modality-aligned representation accessible to current vision-language models.
>
> Nevertheless, we appreciate the suggestion and will highlight this as a direction for extending VLM compatibility with audio tasks.
>
> -------
> [1] Ma, Yinghao, et al. "CMI-Bench: A Comprehensive Benchmark for Evaluating Music Instruction Following." arXiv preprint arXiv:2506.12285 (2025).

---

### Official Review · Reviewer_dwrs · 2025-07-05

**Rating:** 4
**Confidence:** 4

**Summary:**

This paper presents OmniBench, a new multimodal benchmark designed to evaluate general reasoning abilities of large language models across three major modalities: image, audio, and text. The authors also introduce OmniInstruct, a fine-tuning dataset, and conduct extensive evaluation across a variety of open and proprietary models using settings that include raw multimodal inputs and textual approximations. The paper further includes human evaluations, hypothesis testing, and modality-specific analysis.
﻿
While the benchmark provides a reasonably comprehensive evaluation of model behavior across modalities, the benchmark construction methodology and the evaluation logic are not well aligned, and several of its conclusions are not well-supported by the current experimental setup. Although the benchmark itself is potentially valuable, the work lacks sufficient depth in reasoning modeling, and does not yet offer a clear paradigm-shifting contribution to the field.

**Dataset Code Accessibility:**

Partly

**Dataset Code Comments:**

Although the testing pipeline of OmniBench is well-designed and comprehensive, the usability and reproducibility of the OmniInstruct dataset remain limited. While the dataset is publicly accessible, its usage is not well-documented. The paper lacks essential details such as the fine-tuning configuration, training workflow, and replication scripts. This significantly hinders the reproducibility of the reported results and poses challenges for external researchers who may wish to adapt the dataset or benchmark for their own multimodal models.

**Ethical Considerations:**

No, there are no or only very minor ethics concerns

**Final Justification:**

After reading the rebuttal, I believe that several of my main concerns have been adequately addressed. The authors clarified the annotation process and human evaluation design, which resolved my doubts about annotator expertise and the validity of certain benchmark questions. They also provided more concrete examples to support claims of tri-modal complementarity and clarified the types of reasoning involved. Although the benchmark remains limited in scale and does not explicitly model hierarchical or causal reasoning, it presents a timely and well-structured contribution to multimodal evaluation.

**Limitations Weaknesses:**

1.Although the paper emphasizes the importance of vision-audio complementarity and multimodal reasoning, its evaluation primarily focuses on comparing performance across modality-dropping settings (e.g., missing image/audio). The analysis fails to explicitly demonstrate how models leverage complementary modality signals to enhance reasoning.

2.The benchmark does not incorporate tasks or analysis that capture hierarchical or causal reasoning paths across modalities. As such, many tasks seem closer to classification or recognition, rather than probing true multimodal inference.

3.Only three annotators were involved, with limited domain expertise (e.g., music or linguistics). Moreover, several test questions could not be confidently answered by humans, raising questions about whether some benchmark tasks are ambiguous or ill-defined, rather than merely difficult.

4.While many findings are statistically validated, the paper provides little insight into the architectural or representational causes behind the observed performance gaps, limiting its utility for guiding model improvements.

5.While the benchmark claims to target “omni-language models,” its scope is currently limited to three modalities. There's no clear discussion of extending the benchmark to other modalities, despite the terminology implying broader applicability.

6.The OmniBench only contains 1142 samples, yet spans diverse task types. It remains unclear whether this small test set is sufficiently representative to support generalizable conclusions across modalities.

**Strengths Contributions:**

1.The benchmark addresses a timely and important challenge—evaluating LLMs across vision, audio, and text modalities—aligned with the field’s growing interest in generalist models.

2.The study includes comprehensive comparisons between models, modality ablations, and even statistical significance testing, which contributes to a clearer understanding of model behavior.

3.The paper follows NeurIPS guidelines on fair annotation, uses multilingual data, and includes considerations of dataset fairness and representativeness.

4.Both OmniBench and OmniInstruct are accessible for public, adding practical utility for the research community.

---

> ### Author Rebuttal · Authors · 2025-07-31
>
> **We thank the reviewer for the detailed feedback and for acknowledging the benchmark’s importance, annotation protocol, statistical rigor, and potential impact. We appreciate the opportunity to clarify the evaluation design, methodological choices, and future direction.**
>
>
> ### **1. On Complementarity and Cross-Modal Reasoning**
>
> We appreciate the concern regarding the clarity of modality complementarity. In our revision, we will highlight *tri-modal examples where correct predictions only occur when all three modalities are present*—specifically, cases where removing either audio or vision degrades performance, demonstrating the need for complementary information. For example, some questions require detecting emotional mismatch between visual scenes and musical tone, or reasoning about object identity through sound-image alignment. We will add a table (Appendix C) showcasing such cases.
>
> While some tasks may resemble recognition or classification, the **minimum required operation is often multi-step reasoning**, such as mapping instrument cues to context, counting temporal events, or interpreting conflicting modalities. We note that benchmarks like BIG-Bench [1] or HellaSwag [2] also use classification formats to probe deep reasoning. We believe format alone does not preclude cognitive depth.
>
> ### **2. On Reasoning Depth and Task Types**
>
> OmniBench includes reasoning tasks involving:
>
> - **Symbolic logic** (e.g., matching audio cues to text clues), \
> **Causal inference** (e.g., predicting consequence given sound and scene),
>
> - **Quantitative reasoning** (e.g., counting musical or visual patterns), \
> **Commonsense reasoning** (e.g., emotional mismatch or scene contradiction).
>
> We also highlight instances of **spatial-imagery reasoning** (e.g., interpreting spatial layout from soundstage cues). While our current benchmark avoids explicit chain-of-thought traces, we believe it probes deep representational understanding across modalities. A clearer task taxonomy will be added in the revision (Section 3.1 and Appendix A).
>
> ### **3. On Annotation Quality and Human Evaluation**
>
> As the expression of “annotator” could be ambiguous, we respectfully clarify several points:
>
> - Question creation involved **16 annotators and 5 quality reviewers** as stated in sec 3.2, covering music, linguistics, phonetics, cryptography, and math.
>
> - During verification, annotators were required to document the **modality-specific rationale** for each question. Verifiers explicitly assessed this rationale for consistency. This supports our claim that questions are difficult, but **not ill-defined**.
>
> - The final human test was reviewed by **evaluators with X-minute answer time limit (presented in the draft) and another set of experts with more time permit (2-10 minutes), deriving a higher score of 74.03%**. This includes the comparison between humans with strict and loose time limits to acquire responses to estimate difficulty vs. ambiguity.
>
> We will clarify these methodological steps and human evaluation statistics in the revised Appendix B.
>
> ### **4. On Representational Gaps and Model Architecture**
>
> Our goal was to identify representational challenges rather than propose new architectures. Nonetheless, our findings suggest design implications:
>
> - Models like **Qwen-Omni**, which fuse independently tuned vision and audio models, outperform those with static encoders.
>
> - Our results support the hypothesis that models like **GPT-4o and Gemini** may benefit from **unified tokenization and modality alignment** (e.g., AnyGPT-style architectures), rather than modality-specific modules like Q-formers.
>
> We will add this discussion on *representational alignment and architecture implications* to Section 5
>
> ### **5. On the Scope of “Omni” and Other Modalities**
>
> We acknowledge that the term “omni-language models” may imply more than three modalities. Our choice to focus on **image, audio, and text** reflects their foundational role across vision, speech, music, and environmental scenes. These three modalities already challenge alignment, reasoning, and generation, and are extensible to downstream domains such as:**Medical AI** (e.g., sonography + reports), **Finance** (e.g., charts + spoken instructions), **Biosignals** (e.g., EEG + textual prompts), **Structured data** (e.g., MIDI scores as text + acoustic renderings). We view this as a **scalable core** rather than a limitation. We will revise Section 1 to reflect this.
>
> ### **6. On Sample Size and Statistical Generalization**
>
> Though the benchmark contains 1,142 questions, they are **intentionally diverse**, and we argue the **inter-sample heterogeneity** justifies its use for statistical comparison. Appendix D.1 outlines our **bootstrapped hypothesis testing framework**, which supports generalizable model comparisons *within task types*.
>
> To draw a parallel: standardized complex reasoning tests math like **AIME** use ~30 high-difficulty items to evaluate reasoning under constraints. The size of a benchmark should be defined not solely by volume, but by *diversity, precision, and statistical power*. As the reviewer notes, our tasks already span a wide range—contradicting the earlier claim that they lack diversity.
>
> We will clarify this apparent inconsistency and strengthen our justification in Section 3.4 and Appendix D.
>
> ### **7. On OmniInstruct Reproducibility**
>
> We acknowledge that our current documentation for **OmniInstruct** fine-tuning can be improved. However,we need to clarify that the straining checkpoints are all publicly available and not special training tricks are introduced. In the revision, we will:
>
> - Release training scripts (LoRA + full fine-tuning),
>
> - Provide hyperparameter settings and compute budgets,
>
> - Add logs and checkpoints for baseline models.
>
> This will enhance reproducibility and lower the barrier for other researchers to leverage OmniInstruct effectively.
>
>
> ------
> [1] Srivastava, Aarohi et al. “Beyond the Imitation Game: Quantifying and extrapolating the capabilities of language models.” ArXiv abs/2206.04615 (2022): n. pag.
>
> [2] Zellers, Rowan et al. “HellaSwag: Can a Machine Really Finish Your Sentence?” Annual Meeting of the Association for Computational Linguistics (2019).

---

> > ### Comment · Reviewer_dwrs · 2025-08-04
> > **Thanks for the rebuttal**
> >
> > I appreciate your detailed response, which satisfactorily addressed my concerns. I will increase the score accordingly.

---

### Official Review · Reviewer_cpiW · 2025-07-18

**Rating:** 4
**Confidence:** 2

**Summary:**

The paper introduces OmniBench, a novel benchmark designed to evaluate multimodal large language models (MLLMs) on their ability to process and reason across visual (image), acoustic (audio), and textual inputs simultaneously. The authors define models capable of such tri-modal processing as omni-language models (OLMs). OmniBench is distinguished by its high-quality human annotations and the requirement that accurate responses necessitate integrated understanding across all three modalities.

**Dataset Code Accessibility:**

Partly

**Ethical Considerations:**

No, there are no or only very minor ethics concerns

**Final Justification:**

The authors' rebuttal addresses my major concerns. I decide to keep my score.

**Limitations Weaknesses:**

1. With only 1,142 samples, the benchmark may not fully capture the diversity of tri-modal tasks. The authors acknowledge this and plan to expand it in future work.

2. The benchmark does not include video inputs, limiting its applicability to dynamic, temporally aligned multimodal scenarios.

3. Line 137. Missing references.

**Strengths Contributions:**

1. OmniBench fills a critical gap in evaluating tri-modal understanding, as existing benchmarks primarily focus on dual-modality tasks (e.g., image-text or audio-text).

2. The annotation protocol ensures high-quality data, with multiple inspection stages (human and model-assisted) to enforce the requirement that correct answers depend on all modalities.

3.  The paper evaluates a wide range of models, including open-source OLMs, VLMs, ALMs, and proprietary APIs, providing a thorough analysis of current capabilities and limitations.

4. The analysis reveals biases (e.g., models perform better on speech than sound events or music) and highlights the challenges of modality integration, offering clear directions for future work.

---

> ### Author Rebuttal · Authors · 2025-07-31
>
> We sincerely thank the reviewer for their thoughtful assessment and for highlighting the significance of OmniBench in advancing tri-modal evaluation of large language models. We appreciate the opportunity to clarify the key concerns raised.
>
> ### **1. Benchmark Scale and Diversity**
>
> Although a larger dataset would better capture the breadth of tri-modal reasoning, we emphasize that **we prioritized diversity and significance over quantity in this initial release**. Each question was designed to probe distinct reasoning types, minimizing redundancy. Furthermore, we ensured analytical rigor by conducting **statistical significance testing** between model groups (see Appendix C), ensuring that our reported performance trends are meaningful and reproducible despite the dataset size. We have already begun expanding OmniBench with both automated and human-in-the-loop pipelines and plan to release a larger version in follow-up work.
>
> ### **2. Lack of Video Inputs**
>
> While **video data naturally encodes richer temporal dynamics**, our current focus was to study *targeted cross-modal reasoning* using static images and audio. This choice enables **finer control over modality composition and ambiguity**—images can be selectively edited, cropped, or paired with audio to emphasize specific reasoning tasks (e.g., conflicting cues or missing information). At the time of benchmark development, **only GPT-4o and Gemini offered reliable video+audio capabilities**, whereas most open-source video models lacked support for audio streams, making it difficult for the current version of benchmark to include video-audio reasoning. The future work can include synchronized video-audio-text settings in future work.
>
> ### **3. Missing Reference on Line 137**
>
> Thank you for spotting this. The reference on Line 137 was intended to point to a comparative results table included in the Appendix (attatched below). We will fix the in-text citation and ensure that all references are properly included in the camera-ready version.
>
> | Statistic | Current Action & Activity | Story Description | Plot Inference | Object Identification & Description | Contextual & Environmental | Identity & Relationship | Text & Symbols | Count & Quantity | Overall |
> |---|---|---|---|---|---|---|---|---|---|
> | Quantity |  |  |  |  |  |  |  |  |  |
> | Total | 251 | 230 | 237 | 211 | 141 | 32 | 25 | 15 | 1142 |
> | Speech | 78 | 182 | 179 | 162 | 104 | 31 | 22 | 13 | 771 |
> | Sound Event | 147 | 27 | 37 | 28 | 25 | 1 | - | - | 265 |
> | Music | 26 | 21 | 21 | 21 | 12 | - | 3 | 2 | 106 |
> | Word Length |  |  |  |  |  |  |  |  |  |
> | Question | 4.68 | 5.75 | 7.47 | 7 | 6.85 | 6.22 | 7.32 | 8.72 | 6.25 |
> | Option | 8.85 | 7.77 | 8.92 | 6.47 | 5.68 | 10.38 | 11.22 | 6.6 | 8.81 |
> | Img. Rationale | 18.27 | 19.62 | 24.4 | 24.94 | 18.34 | 22.69 | 24.8 | 29.16 | 21.19 |
> | Audio Rationale | 23.11 | 20.5 | 24.4 | 20.97 | 18.27 | 24.92 | 23.1 | 53.84 | 22.9 |
> | Audio Content | 13.21 | 17.91 | 29.87 | 28.03 | 14.41 | 19.01 | 23.31 | 35.16 | 18.37 |
> | Multimodal Info. |  |  |  |  |  |  |  |  |  |
> | Img. Width | 1283.75 | 1291.6 | 2394.93 | 1430.03 | 1141.39 | 1395.53 | 1338.51 | 1787.36 | 1322.36 |
> | Img. Height | 842.32 | 776.11 | 2089.93 | 799.47 | 728.06 | 840.15 | 761.58 | 1168.04 | 818.64 |
> | Audio Len. (s) | 7.35 | 9.82 | 11.22 | 11.43 | 8.03 | 8.63 | 11.43 | 15.63 | 9.22 |

---

> > ### Comment · Reviewer_cpiW · 2025-08-05
> > **Reply.**
> >
> > I appreciate your detailed response, which addresses my major concerns.  I will keep my score.

---

### Note · Authors · 2025-08-15

We wish to extend our sincerest gratitude to the Area Chair and all the reviewers for their thorough evaluation and insightful feedback on our manuscript, "OmniBench: A Novel Benchmark for Tri-Modal Large Language Models." We are encouraged that the reviewers recognized the novelty and significance of our contribution to the field. The constructive comments have been invaluable, and we are confident that addressing them will substantially improve the quality and clarity of our work.

The major points we have addressed and will incorporate into the camera-ready version are summarized as follows:

On Benchmark Design and Scope: We will enhance the paper by providing a detailed breakdown of OmniBench’s category distribution and the design rationale behind our taxonomy. We will also strengthen the justification for our benchmark's scale and scope, emphasizing its intentional diversity and statistical power for evaluating core tri-modal reasoning capabilities. (Addressed for Reviewers cpiW, dwrs, 9aYp).

On Evaluation and Analysis: The revised version will include a more in-depth analysis of cross-modal complementarity, showcasing specific examples where tri-modal inputs are essential for correct reasoning. We will also expand our discussion on the architectural implications of our findings to better guide future model development. (Addressed for Reviewers dwrs, f9w2).

On Annotation and Human Evaluation: We will clarify our multi-stage annotation process, the expertise of our annotators, and the detailed protocol for our human evaluation to underscore that the benchmark's difficulty stems from complex reasoning rather than ambiguity. (Addressed for Reviewers dwrs, f9w2).

On Reproducibility: To ensure the full reproducibility and utility of our work for the community, we will release detailed training scripts and hyperparameters for fine-tuning with the OmniInstruct dataset. (Addressed for Reviewer dwrs).

On Paper Clarity: We will address all minor issues, including correcting the missing reference and clarifying our methodology for handling non-transcript audio, ensuring the final manuscript is polished and precise. (Addressed for Reviewers cpiW, 9aYp).

Thank you once again for your valuable guidance and for helping us improve our paper. We are committed to incorporating all the suggested revisions for the final version.

---

### Decision · Program_Chairs · 2025-09-18

**Decision:**

Accept (poster)

**Comment:**

The paper introduces a novel benchmark for evaluating multimodal large language models (MLLMs) on their ability to process and reason across visual, acoustic, and textual inputs simultaneously, a timely and important topic given the rapid development of MLLMs. Reviewers find that the benchmark addresses a critical gap in evaluating tri-modal understanding. The annotation protocol ensures high-quality data, with multiple inspection stages (human and model-assisted) to enforce the requirement that correct answers depend on all modalities.

The analysis reveals biases (e.g., models perform better on speech than on sound events or music) and highlights the challenges of modality integration, offering clear directions for future research. The study includes comprehensive model comparisons, modality ablations, and statistical significance testing, contributing to a deeper understanding of model behavior. Experiments cover more than 20 models across multiple settings (full tri-modal, ablated modalities, text approximations), providing valuable insights into current limitations. The paper is also well-written and clearly presented.

During the rebuttal, the authors clarified several points and provided a thoughtful summary of areas for improvement through final remarks, which will strengthen revised version.

===== FINAL UPDATE FROM DB Track PCs ====

The final decision for this paper has been taken by the program chairs after consultation with the SACs. All Senior Area Chairs have ranked papers according to the feedback from the AC during the review process. We decided to leave the original meta-review to reflect the opinion of the AC in light of the initial discussions with reviewers and SAC.